# Characteristics of Warm Clouds and Precipitation in South China during the Pre-Flood Season Using Datasets from a Cloud Radar, a Ceilometer, and a Disdrometer

**Jiafeng Zheng [1,*]** , **Liping Liu [2]**, **Haonan Chen [3]**, **Yabin Gou [4]**, **Yuzhang Che [1,5]**, **Haolin Xu [1]** and **Qian Li [1]**

[1] Plateau Atmosphere and Environment Key Laboratory of Sichuan Province, School of Atmospheric Sciences, Chengdu University of Information Technology, Chengdu 610225, China; cyz@cuit.edu.cn (Y.C.); 3170101064@cuit.edu.cn (H.X.); 3170101067@cuit.edu.cn (Q.L.)

[2] State Key Lab of Severe Weather, Chinese Academy of Meteorological Sciences, Beijing 100081, China; liulp@cma.gov.cn

[3] NOAA/Earth System Research Laboratory, Boulder, CO 80305, USA; haonan.chen@noaa.gov

[4] Hangzhou Meteorological Bureau, Hangzhou 310051, China; 2017011201@cuit.edu.cn

[5] Department of Mechanical Engineering, Tokyo Institute of Technology, Ookayama, Meguro-ku, Tokyo 152-8550, Japan

[*] Correspondence: zjf1988@cuit.edu.cn

**Abstract:** The millimeter-wave cloud radar, ceilometer, and disdrometer have been widely used to observe clouds and precipitation. However, there are some drawbacks when those three instruments are solely employed due to their own limitations, such as the fact that radars usually suffer from signal attenuation and ceilometers/disdrometers cannot provide measurements of the hydrometeors of aloft clouds and precipitation. Thus, in this paper, we developed an integrated technology by combining and utilizing the advantages of three instruments together to investigate the vertical structure and diurnal variation of warm clouds and precipitation, and the raindrop size distribution. Specifically, the technology consists of appropriate data processing, quality control, and retrieval methods. It was implemented to study the warm clouds and precipitation in South China during the pre-flood season of 2016. The results showed that the hydrometeors of warm clouds and precipitation were mainly distributed below 2.5 km and most of the rainfall events were very light with a rain rate less than 1 mm h$^{-1}$, however, the stronger precipitation primarily contributed the accumulated rain amount. Furthermore, a rising trend of cloud base height from 1000 to 1900 BJT was found. The cloud top height and cloud thickness gradually increased from 1200 BJT to reach a maximum at 1600 BJT (Beijing Standard Time, UTC+8), and then decreased until 2000 BJT. Also, three periods of the apparent rainfall on the ground of the day, namely, 0400–0700 BJT, 1400–1800 BJT, and 2300–2400 BJT were observed. During three periods, the raindrops had wider size spectra, higher number concentrations, larger rain rates, and higher water contents than at other times. The hydrometeor type, size, and concentration were gradually changed in the vertical orientation. The raindrop size distributions of warm precipitation in the air and on the ground were different, which can be expressed by $\gamma$ distributions $N(D) = 1.49 \times 10^4 D^{-0.9484} \exp(-6.79D)$ in the air and $N(D) = 1.875 \times 10^3 D^{0.862} \exp(-2.444D)$ on the ground, where D and N(D) denote the diameter and number concentration of the raindrops, respectively.

**Keywords:** warm clouds and precipitation; cloud radar; ceilometer; disdrometer; South China

## 1. Introduction

South China is one of the moistest regions of mainland China with a tropical–subtropical climate. Under the influences of the pacific subtropical high, East Asian monsoon, and other synoptic systems, precipitation events frequently occur in this region during the pre-flood season from April to June, accounting for 40%–50% of the annual rain amount [1]. Despite the deep convective and stratiform precipitation contributing to most of the disastrous weather and rainfall in this region, warm clouds and precipitation have a much higher occurrence in the sky due to the abundant water vapor in the lower atmosphere [2,3]. Therefore, they play key roles in the local hydrological cycle, energy budget, and climate. The detection and study of warm clouds and precipitation are valuable for improving our understanding of the related physical and dynamic issues of clouds in this region. They can also provide necessary information for improving the capability of local numerical weather and climate models and help us to conduct realistic missions of weather modification.

Recently, radiosonde, aircraft, satellite, and ground-based remote sensing techniques have been used to study clouds and precipitation in different regions [4–7]. A radiosonde is usually used to obtain in-situ measurements of temperature, relative humidity, wind, and pressure profiles. These measurements can be utilized to estimate the cloud base height, cloud top height, cloud layer number, and cloud thickness as balloons penetrate the cloud layers [8–11]. Nevertheless, the number of radiosonde stations is relatively limited, and a radiosonde cannot provide the specific parameters of hydrometeors [12,13]. Aircraft penetration typically yields a detailed description of the horizontal structure of clouds and precipitation and can obtain the hydrometeor properties in the vertical direction by making multiple passes at different height levels [14,15]. This is a reliable way to simultaneously obtain high-quality microphysical hydrometeor measurements, such as particle size, particle number concentration, particle phase, and water content [16,17]. However, aircraft penetration is very costly and can only provide instantaneous measurements during a specific period and over particular regions [18]. Satellites equipped with remote sensors have significant advantages regarding spatial coverage. They can provide data without any topographic limitations. For instance, geostationary satellites, such as the Himawari series, geostationary operational environmental satellite series, Fengyun-2 (FY-2) series, and Fengyun-4A (FY-4A) can offer real-time cloud maps over fixed areas with large regional coverages and provide multispectral measurements, including cloud top height, cloud temperature, cloud types, and cloud movement [19–22]. Nonetheless, the spatial and temporal resolutions of geostationary satellites are still relatively low and can hardly satisfy the requirements of fine-scale observation and research of clouds and precipitation [2]. Polar-orbiting satellites equipped with meteorological radars are more prevalently used in clouds and precipitation vertical detection, such as the tropical rainfall measuring mission, Cloudsat, cloud-aerosol lidar and infrared pathfinder satellite observations, and global precipitation measurement [23–25]. These on-board radars can profile the nether cloud layers as satellites moving in orbit, and obtain precise measurements, such as radar reflectivity, hydrometeor type, particle number concentration, water content, and rain rate. [26–30]. These measurements have been widely used in many studies for evaluating the cloud radiative effect and influence on climate, for investigating the features of high-impact severe weather systems, and for revealing the physical features and vertical structures of different clouds and precipitation types [31–34]. However, satellite-based radars possess limitations for continuous observation of shallow clouds and precipitation because of the limited number of times per day and the contamination by surface clutter at low altitudes [35]. In the continental region, ground-based remote sensors, such as millimeter-wave cloud radar (MMCR), laser ceilometer (CL), and disdrometer are better alternatives for long-term in-situ observation of clouds and precipitation. Many studies have indicated that the MMCR has a good ability to detect different cloud types from weak clouds to relatively strong precipitating clouds, such as cirrus, stratocumulus, cumulus, low-level stratiform clouds, etc., because of its high sensitivity [36–39]. By operating in the vertically pointing mode with the transmission of narrow pulses, MMCR also has very high spatial and temporal resolutions, which are of only a few decameters and seconds, respectively. Thus, the radars can continuously profile aloft clouds and precipitation in a short-time interval and

provide subtle vertical observations. MMCRs have been used in many significant science programs in many countries, such as the Atmospheric Radiation Measurement projects in the USA, the Third Tibetan Plateau Atmospheric Science Experiment in China, and the Cloudnet in Europe [40–42]. However, MMCR is generally deployed in conjunction with other instruments, such as with a CL and disdrometer, because it suffers from signal attenuation and is incapable of detecting the cloud base height under precipitating conditions. A CL and disdrometer can complement more reliable and sophisticated measurements of cloud base height and raindrop spectra information [43,44]. Whereas, inversely, they cannot provide measurements of hydrometeors of aloft cloud and precipitation. Overall, the MMCR, CL, and disdrometer both have individual limitations for clouds and precipitation detection. A comprehensive usage of these remote sensors will be a more appropriate way for the thorough observation and study of cloud and precipitation aloft and on the ground.

For the warm clouds and precipitation in South China, several studies using aircraft and radar measurements have been presented. For instance, in 1988, Wu et al. first investigated the properties of warm stratocumulus and cumulus clouds during the pre-flood season using aircraft measurements. They proposed that the water content of stratocumulus clouds in this region is much larger than the counterpart in Northern China; cumulus clouds have similar features as those over the ocean; many cloud droplets have diameters greater than 40 μm, and the collision and coalescence processes are the main mechanisms of precipitation formation [45]. In 2012, Ma et al. further studied stratocumulus clouds in the winter using aircraft and found that aloft warm clouds have a multiple-layered structure with small space intervals. The precipitating clouds possess higher cloud base heights and top heights than non-precipitating clouds. The mean values for the number concentration, diameter, and water content of hydrometeors in stratocumulus clouds were determined to be 652 cm$^{-3}$, 18.2 μm, and 1.03 g·m$^{-3}$, respectively [46]. In 2016, Liu et al. merged the data of the Ka-MMCR, C-band profiling radar, and CL to investigate the vertical structure and diurnal cycle of the entire clouds and precipitation events in South China. They proposed that there were three periods of high occurrence frequency of low-level clouds that occurred at sunrise, noon, and sunset. Large amounts of clouds were concentrated below 3 km [2]. Huo et al. studied the raindrop size distributions of different rain types in this region using a disdrometer, and found that shallow precipitation typically yields an uneven distribution for the ground rain rate [3]. Despite that some conclusions of warm clouds and precipitation in South China have been proposed as mentioned above, the specific aspects concerned with the vertical structure, diurnal cycle, and physical property of warm clouds and precipitation during the pre-flood season have not yet been investigated.

Motivated by the two aspects as mentioned above (the limitations of individual remote sensor and the lack of thorough study on warm clouds and precipitation in South China during the pre-flood season), in this study, we attempted to develop an integrated technology using three different measuring technologies (the Ka-band MMCR, CL, and disdrometer) for an investigation of the vertical structure and diurnal variation of warm clouds and precipitation, and the raindrop size distribution. The technology was based on a mix of appropriate and standard data processing, quality control and retrieval methods. This technology was applied to elucidate the warm clouds and precipitation in South China during the pre-flood season in 2016. The remainder of the paper is organized as follows. Section 2 presents detailed descriptions of the three instruments, their data processing and quality control (QC) technologies, warm clouds and precipitation determinations, and physical quantity retrieval methods. Section 3 provides the results of data QC and provides an analysis of the general characteristics, diurnal variation, vertical structure, and particle size distribution of warm clouds and precipitation. Section 4 provides a discussion of the differences between our study and previous results. Section 5 ends the paper with a summary.

## 2. Materials and Methods

### *2.1. Instruments and Measurements*

To advance understanding of clouds and precipitation in South China during the pre-flood season, in 2016, the Chinese Academy of Meteorological Sciences carried out an observation experiment in Guangdong province of China. During the experimental period from 14 April to 18 June, the Ka-band MMCR, CL, and disdrometer, were simultaneously deployed at Longmen Weather Observatory (LM, 23.783°N, 114.25°E, 86 m above sea level). The three instruments were expected to provide continuous, long-term, and high-resolution measurements of clouds and precipitation over the site and on the ground. The experimental location and equipment are shown in Figure 1.

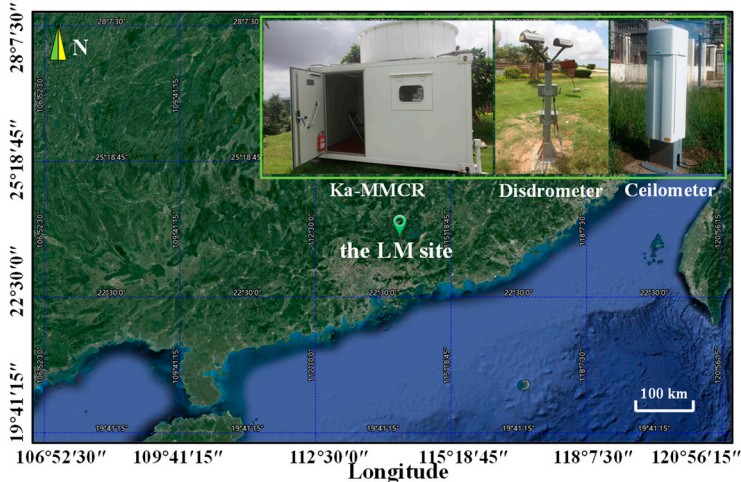

**Figure 1.** Experimental location of the Longmen Weather Observatory (LM, 23.783°N, 114.25°E, 86 m above sea level) and Ka-MMCR, disdrometer, and ceilometer (CL) instruments.

### 2.1.1. Ka-Band MMCR

The Ka-band MMCR is a Doppler, solid-state and polarimetric radar. It works at 33.44 GHz, with a wavelength of 8.9 mm and a peak power over 100 W. By operating in vertically pointing mode, the radar can continuously observe vertical profiles of the Doppler spectrum (SP), radar reflectivity (Z, dBZ), mean Doppler velocity ($V_M$, m·s$^{-1}$), spectrum width ($S_w$, m·s$^{-1}$), and linear depolarization ratio (LDR, dB) of the loft clouds and precipitation in a height range from 0.15 to 15.3 km with a spatial resolution of 30 m and a temporal resolution of ~9 s. To meet the requirement of clouds and precipitation observations at different heights with different intensities, multiple radar operational modes were designed by configuring with different radar parameters and signal processing technologies. Detailed descriptions of their performances and differences can be found in a previous study [2]. Herein, for the sake of the study of warm clouds and precipitation, measurements observed by radar precipitation mode were used. This mode possesses a much smaller blind range and wider reflectivity and Doppler velocity ranges than the counterparts of the other modes. However, it is worth noting that this mode is less sensitive, as a result, a part of cloud droplets with diameters less than ~0.12 mm is inevitably unavailable. Table 1 presents the main technical parameters of the MMCR operating in the precipitation mode.

**Table 1.** Main technical parameters of the MMCR (millimeter-wave cloud radar) operating in the precipitation mode.

| No. | Items | Technical Specifications |
|:---:|---|---|
| 1 | Frequency (Wavelength) | 33.44 GHz (8.9 mm) |
| 2 | Transmitted peak power | ≥100 W |
| 3 | Beam width | 0.3 degree |
| 4 | Pulse repetition frequency | 8333 Hz |
| 5 | Gate number | 510 |
| 6 | Vertical resolution | 30 m |
| 7 | Horizontal resolution | 26 m at 5 km |
| 8 | Temporal resolution | ~9 s |
| 9 | Transmitted pulse width | 0.2 μs |
| 10 | Spectrum bin number | 256 |
| 11 | Detectable height range | 0.15–15.3 km |
| 12 | Detectable reflectivity range | −33–30 dBZ |
| 13 | Detectable velocity range | $-18.67$–$18.67$ m·s$^{-1}$ |
| 14 | Spectrum velocity resolution | 0.145 m·s$^{-1}$ |
| 15 | Measurements | Doppler spectrum, reflectivity, mean Doppler velocity, spectrum width, and linear depolarization ratio |

### 2.1.2. Ceilometer and Disdrometer

The CL was made by Vaisala Company (Vantaa, Finland) and designed using pulsed diode laser light detection and ranging (LIDAR) technology. It emits a laser pulse in the vertical direction and receives the backscattered signal reflected from the cloud, precipitation, or other targets. The CL provides a backscatter profile and cloud base height with a vertical resolution of 10 m and a temporal resolution of 2–3 s. The disdrometer used was a Parsivel disdrometer (PARSIVEL) made by OTT Hydromet Company (Kempten, Germany). It is equipped with a laser-optical transmitter that can simultaneously detect the particle size and falling speed based on signal attenuation caused by the passing hydrometeor. The measuring height was 1.4 m above the ground, and the sampling time was 60 s. The diameter and velocity information of hydrometeors were recorded by division into 32 non-equidistant classes. Moreover, the PARSIVEL can also provide rainfall quantities, including drop size distribution (DSD), reflectivity (Z, dBZ), rain rate ($R_R$, mm·h$^{-1}$) and rain amount ($R_M$, mm). The main technical parameters of these two instruments are listed in Table 2.

**Table 2.** Main technical parameters of ceilometer (CL) and disdrometer (PARSIVEL) used in this study.

| CL/PARSIVEL | No. | Items | Technical Specifications |
|:---:|:---:|---|---|
| | 1 | Sensor type | Laser, pulsed |
| | 2 | Wavelength | 910 ± 10 nm |
| | 3 | Peak power | 27 W |
| CL | 4 | Sampling volume | $834 \times 266 \times 264$ mm$^3$ |
| | 5 | Detection range | 0–15 km |
| | 6 | Spatial resolution | 10 m |
| | 7 | Temporal resolution | 2 s |
| | 8 | Measurements | Backscatter profiles, cloud base height |

**Table 2.** *Cont.*

| CL/PARSIVEL | No. | Items | Technical Specifications |
|---|---|---|---|
| PARSIVEL | 1 | Sensor type | laser |
| | 2 | Peak power | $\geq$2 W |
| | 3 | Sampling height | 1.4 m |
| | 4 | Sampling area | 54 cm$^2$ |
| | 5 | Measurable diameter range | 0.062–24.5 mm |
| | 6 | Measurable velocity range | 0.05–20.8 m$\cdot$s$^{-1}$ |
| | 7 | Temporal resolution | 60 s |
| | 8 | Measurements | Drop size distribution, reflectivity, rain rate, rain amount, weather code |

### 2.2. Data Processing, Quality Control (QC), and Physical Quantity Retrieval Methods

Previous studies indicated that data quality issues can affect the application of Ka-MMCR and PARSIVEL [47,48]. Therefore, appropriate technologies of data processing and QC were adopted in this study. Moreover, specific retrieval methods were used to obtain the specific physical quantities for the warm clouds and precipitation. Figure 2 shows a brief flowchart of data processing, QC, and physical quantity retrieval for MMCR and PARSIVEL that we adopted. Eventually, 18 types of measurements and retrievals were produced, including radar quality-controlled SP, Z, $V_M$, $S_W$, and LDR, newly calculated radar spectral skewness ($S_K$), spectral kurtosis ($K_T$), and radar-derived (or PARSIVEL/CL-measured) cloud and precipitation physical quantities, including the cloud base height (CBH, km), cloud top height (CTH, km), cloud thickness (CTK, km), cloud layer number (CLN), $R_R$, $R_M$, DSD, vertical air velocity ($V_A$, m$\cdot$s$^{-1}$), particle mean terminal velocity ($V_T$, m$\cdot$s$^{-1}$), particle mean diameter ($D_M$, mm), particle total number concentration ($N_T$, m$^{-3}$), and liquid water content (LWC, g$\cdot$m$^{-3}$). For convenience, abbreviations used in this manuscript can be found in the Appendix A (Table A1). Each step in Figure 2 is explained in detail in the following subsections.

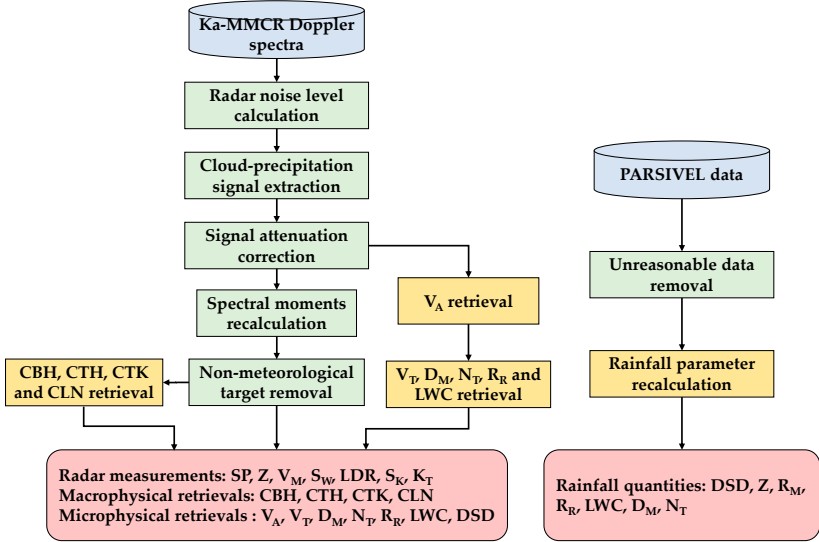

**Figure 2.** Flowchart of the data processing, quality control (QC), and cloud and precipitation physical quantity retrieval for Ka-MMCR (millimeter-wave cloud radar) and PARSIVEL disdrometer. The SP, Z, $V_M$, $S_W$, LDR, $S_K$, and $K_T$ represent radar Doppler spectrum, reflectivity, mean Doppler velocity, spectrum width, linear depolarization ratio, spectral skewness, and spectral kurtosis, respectively; the CBH, CTH, CTK, and CLN denote radar-derived cloud base height, cloud top height, cloud thickness, and cloud layer number, respectively; the $V_A$, $V_T$, $D_M$, $N_T$, $R_R$, $R_M$, LWC, and DSD are vertical air velocity, particle mean falling velocity, particle mean diameter, particle total number concentration, rain rate, rain amount, liquid water content, and drop size distribution, respectively.

### 2.2.1. Ka-MMCR Data Processing, QC, and Physical Quantity Retrieval

The Ka-band MMCR undergoes attenuation as its electromagnetic wave passes through clouds–precipitation. Non-meteorological echo caused by plankton also contaminates radar data in the low-level atmosphere [36]. Moreover, the radar originally provides SP, Z, $V_M$, $S_W$, and LDR, and the other 12 kinds of measurements and retrievals remain to be further produced. Considering this fact, the following techniques were implemented.

(1) Radar noise level calculation. The cloud-precipitation signal is overlapped by radar noise in the Doppler spectrum. For separation, an objective technology proposed by Hildebrand and Sekhon was utilized to estimate radar noise level [49].

(2) Cloud-precipitation signal extraction. All continuous spectral bins above radar noise level were picked out and further judged by a signal-to-noise ratio (SNR) threshold ($\geq -12$ dB) and a bin-number threshold ($\geq 5$) because cloud-precipitation signal typically has a higher power and larger spectral width than radar noise [47,50]. Only consecutive signals with the first two powers were reserved, and their SNRs, left endpoints, right endpoints, and peaks were recorded.

(3) Signal attenuation correction. The radar returned signal is attenuated by hydrometeors, causing underestimations of the measured SP and Z. For correction, an iterative procedure was implemented [47,51].

$$K_i = \alpha Z_c(i)^\beta \tag{1}$$

$$\tau_i = \tau_{i-1} \times \exp(-2 \times K_i \times \Delta R) \tag{2}$$

$$Z_c(i) = \frac{Z_m(i)}{\tau_{i-1}} \times \exp(K_i \times \Delta R) \tag{3}$$

$$SP_c(n,i) = \frac{SP_m(n,i)}{\tau_{i-1}} \times \exp(K_i \times \Delta R) \tag{4}$$

In Equations (1)–(4), $i$ and $n$ denote the radar range gate number and spectral bin number, respectively, $K_i$ (dB·km$^{-1}$) is the attenuation coefficient, $\tau_i$ is the radar wave two-way transmissivity, $\Delta R$ (30 m) is the gate length, $Z_m$ and $Z_c$ represent the radar-measured and corrected reflectivity, respectively, and $SP_m$ and $SP_c$ represent the radar-measured and corrected Doppler spectra, respectively. To start the iteration, the initial $\tau_0$ and $Z_c(0)$ were set to 1 and $Z_m(0)$, respectively. The coefficients $\alpha$ and $\beta$ were set to 0.00334 and 0.73, respectively [52].

(4) Spectral moment recalculation. After attenuation correction of SP, six radar moments including Z, LDR, $V_M$, $S_W$, $S_K$ and $K_T$ were recalculated using the following formulas:

$$P_{c\&p} = \sum_{v=V_l}^{v=V_r} (SP_c(v) - P_N) \tag{5}$$

$$Z = 10 \times log_{10}\left(\frac{P_{c\&p} \times R^2}{C}\right), C = \frac{P_t \times G^2 \times \theta \times \varphi \times h \times \pi^3 \times |k|^2}{1024 \times ln2 \times \lambda^2 \times L_\varepsilon} \tag{6}$$

$$LDR = Z_V - Z_H \tag{7}$$

$$V_M = \frac{\sum_{v=V_l}^{v=V_r} v \times (SP_c(v) - P_N)}{\sum_{v=V_l}^{v=V_r} (SP_c(v) - P_N)} \tag{8}$$

$$S_W = \left[\frac{\sum_{v=V_l}^{v=V_r} (v - V_M)^2 \times (SP_c(v) - P_N)}{\sum_{v=V_l}^{v=V_r} (SP_c(v) - P_N)}\right]^{\frac{1}{2}} \tag{9}$$

$$S_K = \frac{\sum_{v=V_l}^{v=V_r} (v - V_M)^3 \times (SP_c(v) - P_N)}{S_W{}^3 \times \sum_{v=V_l}^{v=V_r} (SP_c(v) - P_N)} \tag{10}$$

$$K_T = \frac{\sum_{v=V_l}^{v=V_r} (v - V_M)^4 \times (SP_c(v) - P_N)}{S_W{}^4 \times \sum_{v=V_l}^{v=V_r} (SP_c(v) - P_N)} - 3 \tag{11}$$

where $v$ denotes the Doppler velocity of the spectral bin, $V_l$ and $V_r$ (m·s$^{-1}$) denote the left-endpoint and right-endpoint Doppler velocities of the cloud–precipitation signal in the spectra, respectively, $SP_c(n)$ (mW) is the signal power of each spectral bin, $P_N$ (mW) is the noise level, $P_{c\&p}$ (mW) represents total power of the cloud–precipitation signal in the spectra, $R$ (km) is the distance from the radar to target, C is the radar constant, $P_t$ (W) is the radar transmitted power, $G$ (dB) is the antenna gain, $\theta$ and $\varphi$ (degree) are the radar horizontal and vertical beam widths, respectively, $h$ (km) represents the spatial pulse length, $\lambda$ (mm) is the radar wavelength, $|k|^2$ is the refractive index, $L_\varepsilon$ (dB) is the feeder loss, and $Z_H$ and $Z_V$ (dBZ) are the two reflectivities received by the radar parallel and cross-polarization channels.

(5) Non-meteorological echo removal. Non-meteorological echo in MMCR caused by low-level plankton, which consists of dust, insects, pollen, and other targets, were commonly observed in the low- and mid-latitude regions [53,54]. MMCR-measured Z can be used in conjunction with CL-measured CBH to identify and remove the plankton echo [2]. However, this approach cannot remove the entire plankton echo because part of the plankton actually exists above the CBH. In this study, we used a simple technology called the "Z-LDR double-threshold" to eliminate the plankton contamination in the MMCR data [55]. This method is based on the observational fact that the Z and LDR distributions of plankton and warm clouds and precipitation are apparently different. Specifically, the plankton echo can exhibit a very large LDR with a relatively small Z. In contrast, the cloud and precipitation echo generally have a relatively small LDR with a wide range of Z. According to the realistic statistical result from MMCR data (as shown in Figure 3), the Z and LDR thresholds were simultaneously set to −8 dBZ and −14 dB, respectively. In this case, any radar range gate that simultaneously possesses a Z smaller than −8 dBZ and an LDR larger than −14 dB can be judged as plankton and be removed. Using this "Z-LDR double-threshold", all plankton echo in the LDR field can be fully filtered out as expected, whereas a part of the scattered plankton will remain in other radar moments, which have a larger echo amount than LDR. Therefore, a $3 \times 3$ filtering window was further implemented for Z, $V_M$, $S_W$, $S_K$, and $K_T$ to eliminate the remain scattered plankton [55].

(6) Retrieval of the cloud-precipitation macrophysical quantity. The CBH, CTH, CTK, and CLN of cloud and precipitation were derived by using the radar-measured Z. For each radar radial, continuous segments with more than ten gates (300 m) with radar available Z were distinguished and the segment base height and top height were taken as CBH and CTH, respectively. The segment number and length were regarded as CLN and CTK, respectively.

(7) Retrieval of the cloud-precipitation microphysical parameters. Seven key microphysical parameters of warm clouds and precipitation, including $V_A$, $V_T$, $D_M$, $N_T$, $R_R$, LWC, and DSD were further deduced using the processed radar Doppler spectra. First, a technology called "small-particle-trace" was applied to estimate the $V_A$ from Doppler spectra. This approach has been applied and verified by Gossard, Kollias, Shupe, Zheng, and Sokol in different cloud and precipitation type studies [39,50,56–58]. The $V_T$ was then obtained by subtracting $V_A$ from $V_M$. Thereafter, we shifted the Doppler spectra according to $V_A$ and converted the spectra unit from dBm to dBZ using Equations (5)–(6). The relationship between the particle terminal velocity and diameter must be determined before further retrieval. For the liquid hydrometeor, the relationship can be written as [59,60]:

$$D = \frac{1}{0.6} \times ln\frac{10.3}{9.65 - V_t/\delta(h)} \tag{12}$$

$$\delta(h) = 1 + 3.68 \times 10^{-5} h + 1.71 \times 10^{-9} h^2 \tag{13}$$

where $D$ (mm) and $V_t$ (mm·s$^{-1}$) denote the diameter and terminal velocity of the particle, $h$ (m) is the radar sampling height above sea level, and $\delta(h)$ is a correction factor. Based on this, radar-derived $D_M$, $N_T$, $R_R$, LWC, and DSD can be acquired by using the following formulas [47]:

$$P_{D_i} = \frac{C \times D_i^6}{R^2} \tag{14}$$

$$N(D_i) = \frac{P_i}{P_{D_i} \times \Delta D_i} \tag{15}$$

$$D_M = \frac{\Sigma_{i=D_{min}}^{D_{max}} D_i \times N(D_i) \times \Delta D_i}{\Sigma_{i=D_{min}}^{D_{max}} N(D_i) \times \Delta D_i} \tag{16}$$

$$N_T = \sum_{i=D_{min}}^{D_{max}} N(D_i) \times \Delta D_i \tag{17}$$

$$R_R = \frac{6\pi}{10^4} \sum_{i=D_{min}}^{D_{max}} D_i^3 \times V_t(D_i) \times N(D_i) \times \Delta D_i \tag{18}$$

$$LWC = \frac{\pi}{6000} \sum_{i=D_{min}}^{D_{max}} \rho \times D_i^3 \times N(D_i) \times \Delta D_i \tag{19}$$

where $\Delta D_i$ (mm) is the diameter interval, $P_{D_i}$ (mW) is the power caused by a single particle with a diameter of $D_i$, $P_i$ (mW) is the radar-measured power for the particles with a diameter of $D_i$, $D_{min}$ and $D_{max}$ (mm) represent the detected minimum and maximum diameters in the Doppler spectra, respectively, and $\rho$ (g·cm$^{-3}$) is the water density.

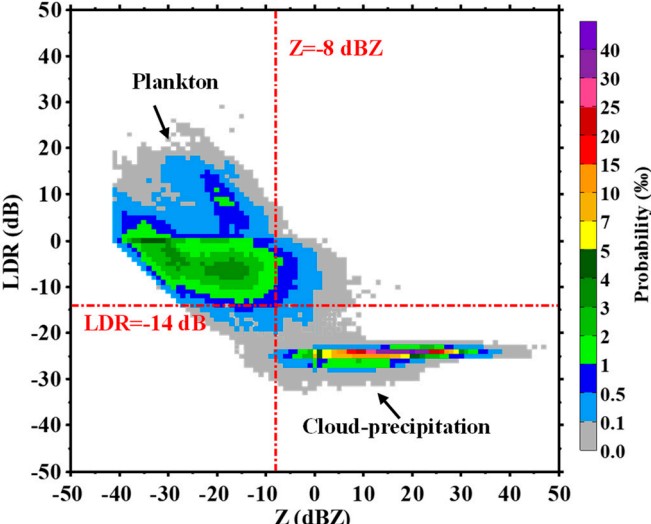

**Figure 3.** Ka-MMCR reflectivity-linear depolarization ratio (Z-LDR) probability distributions for warm clouds and precipitation and plankton. A couple of Z and LDR thresholds were set to −8 dBZ and −14 dB to eliminate the plankton.

### 2.2.2. PARSIVEL Data Processing and QC

Studies indicated that the PARSIVEL disdrometer can produce unreasonable results owing to its inherent limitation [44,61]. The unreasonable data must be removed, and some posterior rainfall quantities need to be recalculated.

(1) Unreasonable data removal. First, considering the system sensitivity and noise, DSDs with a total drop number less than ten and a rain rate smaller than $0.002 \cdot \text{mm} \cdot \text{h}^{-1}$ were removed, otherwise, they were regarded as valid rainy DSDs [44]. Second, any raindrop with a diameter greater than 1 mm that has a normal falling velocity (or diameter) but with an excessively large or small diameter (or falling velocity) is treated as problematic data, which can be produced when a large raindrop or multiple raindrops pass in parallel through the laser beam or can be caused by a strong wind shear or splashing on the instrument surface during rainfall. These kinds of unreasonable data were recognized by comparing the PARSIVEL-measured result with the theoretical $V_T$-D relationship shown in Equations (12)–(13). Any measured result outside ±60% of the relationship was removed [62]. The above-mentioned method was not used for small raindrops with a diameter smaller than 1 mm because most of the disdrometers severely underestimate the small drop concentration as proposed by Thurai et al. [63].

(2) Rainfall quantity recalculation. After step (1), the number concentration of raindrop in different classes can be obtained by the following formula,

$$N(D_i) = \sum_{j=1}^{32} \frac{n_{ij}}{A_i \times \Delta t \times V_j \times \Delta D_i} \tag{20}$$

where $D_i$ (mm) is the diameter for the $i$th class, $\Delta D_i$ is the interval for the $i$th class, $N(D_i)$ ($\text{m}^{-3} \cdot \text{mm}^{-1}$) is the number concentration of raindrops per unit volume with diameters in the interval from $D_i$ to $D_i + \Delta D_i$, $n_{ij}$ represents the raindrop number within size class $i$ and velocity class $j$, $V_j$ ($\text{m} \cdot \text{s}^{-1}$) is the measured falling velocity for velocity class $j$, $A_i$ ($\text{m}^2$) is the effective sampling area of size class $i$, and $\Delta t$ (60 s) is the sampling time. The $A_i$ can be estimated as $0.18 \times \left(0.03 - 0.5D_i \times 10^{-3}\right)$ [44,62]. The other six rainfall physical quantities, including Z, $R_M$, $R_R$, LWC, $D_M$ and $N_T$, were recalculated using Equations (21)–(25).

$$Z = \sum_{i=1}^{32} \sum_{j=1}^{32} D_i^6 \times \frac{n_{ij}}{A \times \Delta t \times V_j} \tag{21}$$

$$R_R = \frac{R_M}{\Delta t} \times 3600 = \frac{6\pi}{10^4} \sum_{i=1}^{32} \sum_{j=1}^{32} D_i^3 \times \frac{n_{ij}}{A \times \Delta t} \tag{22}$$

$$LWC = \frac{\pi}{6000} \sum_{i=1}^{32} \sum_{j=1}^{32} \rho \times D_i^3 \times \frac{n_{ij}}{A \times \Delta t \times V_j} \tag{23}$$

$$D_M = \frac{\Sigma_{i=1}^{32} D_i^3 \times N(D_i) \times \Delta D_i}{\Sigma_{i=1}^{32} N(D_i) \times \Delta D_i} \tag{24}$$

$$N_T = \sum_{i=1}^{32} N(D_i) \times \Delta D_i \tag{25}$$

### 2.3. Warm Cloud and Precipitation Determination and Data Matching

The entire dataset collected from 15 April to 18 June 2016 at the LM site was processed according to the methods mentioned above, and for our study purpose, only warm clouds and precipitation

events were selected. A cloud and precipitation event was determined as a warm event according to its MMCR-derived CTH, which should be lower than a height threshold of the zero-degree layer. According to the daily results on 0800 and 2000 BJT of the nearest radiosonde station (Heyuan, a site 45-km west to the LM site), the zero-degree height of the atmosphere was in the range of 4.2 to 5.1 km during the observation period. Therefore, the height threshold was set to 4.2 km.

After determination, basic information of the warm clouds and precipitation events during the 65-day observation period were counted. We defined an event as the Ka-MMCR continuously capturing the radar valid profiles for at least 5 min. A radar valid profile was distinguished when there was a radar range segment with available reflectivities of more than ten gates (0.3 km). The results indicate the Ka-MMCR has captured 531 events containing 101,428 (17,131 min) radar valid profiles. The precipitating profile (defined as the radar echoes with reflectivities greater than −10 dBZ attached near the ground and denotes the track of the raindrops falling to the ground) number was 50,477 (8502 min), while the non-precipitating profile number was 50,951 (8629 min). According to the radar precipitating profiles, 5317 samples (5317 min) from PARSIVEL were matched. Similarly, according to the valid profiles of cloud and precipitation, 277,595 (17,131 min) samples from CL were matched.

## 3. Results

After data matching, the simultaneous datasets of the three instruments were used to study the following aspects of the warm clouds and precipitation in South China during the pre-flood season. First, evaluations of data QC effects of Ka-MMCR and PARSIVEL are presented in Section 3.1. Subsequently, the measurements of Ka-MMCR and PARSIVEL were combined to investigate the hydrometeor distribution and rainfall general characteristics, as described in Section 3.2. As described in Section 3.3, measurements of the three instruments were comprehensively used to compare and analyze the diurnal variations of clouds and precipitation overhead and on the ground. The vertical structure of clouds and precipitation and the raindrop size distribution are elucidated in Sections 3.4 and 3.5, respectively.

### 3.1. Data QC Result

Plankton targets mainly are present in the planet's boundary layer and can affect the data usage of Ka-MMCR for cloud and precipitation observations. For separation of the plankton contamination from radar dataset, Z-LDR probability distributions of plankton and warm clouds and precipitation were each investigated. The statistic results are shown in Figure 3, which indicate that the distributions of plankton, clouds, and precipitation were obviously different in the Z and LDR fields. Namely, the plankton simultaneously had a smaller Z primarily in the range of −41 to 1 dBZ and a remarkably large LDR within −22 to 18 dB. In contrast, the clouds and precipitation simultaneously had a larger Z from −9 to 38 dBZ and an apparently narrow scope of LDR from −29 to −22 dB. Based on these differences, a couple of thresholds for Z and LDR were set to −8 dBZ and −14 dB by preferentially considering that the warm clouds and precipitation should not be accidentally deleted. Verification demonstrated that 92.22% of the entire plankton targets could be removed from the Ka-MMCR dataset using this Z-LDR threshold combined with a 3 × 3 filtering window; meanwhile, the warm clouds and precipitation echo remains. Figure 4 shows the cumulative probabilities of Z and LDR for the radar dataset before and after QC. Comparing the LDR and Z curves, we found that the LDR was mainly distributed within −27–15 dB before QC, whereas it was concentrated in the range of −27 to −11 dB and had a Z probability lower than −1 dB that decreased after QC because of plankton elimination. Moreover, the result shows that the probability of Z from 0 to 27 dBZ increased after QC because of attenuation correction.

To present the effect of the radar QC, a typical case that contains different types of warm clouds-precipitation and plankton is plotted in Figure 5. There were warm convections, several cumuli, and layers of fracto-cumulus clouds that successively passed over the site in the lower atmosphere during the observation period. The warm convection (1200–1330 BJT) has a relatively large scale

and a high CTH and was hardly surrounded by plankton because of the influences of rain wash and downdraft. The hydrometeors in the convection contribute to radar-measured LDR with a small value range from −26 to −24 dB (Figure 5b). However, after 1330 BJT, scattered plankton targets gradually appeared around the small-scale cumuli and fracto-cumuli. They were mainly located under 1.5 km with an extremely large LDR that was greater than −14 dB. As shown in Figure 5c,d, the scattered plankton targets were eliminated well after QC (as marked by the arrows); meanwhile, the small-scale clouds, such as the weak cumuli and thin fracto-cumuli from 1800 to 2000 BJT remained unchanged. The deviation in Z before and after QC, as shown in Figure 5e, illustrates that most Z of cloud suffers from small attenuation with a value below 0.1 dB. However, for stronger and thicker precipitating cumuli and convection (as marked by the circles), the Z attenuation can be slightly larger than the other clouds with a value within 0.2 and 0.4 dB under the cloud top.

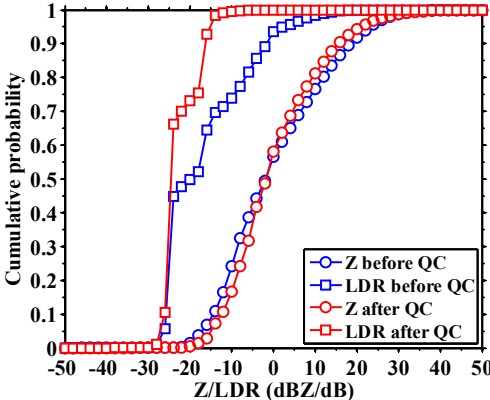

**Figure 4.** Cumulative probability curves of Ka-MMCR reflectivity (Z) and linear depolarization ratio (LDR) before and after data quality control (QC).

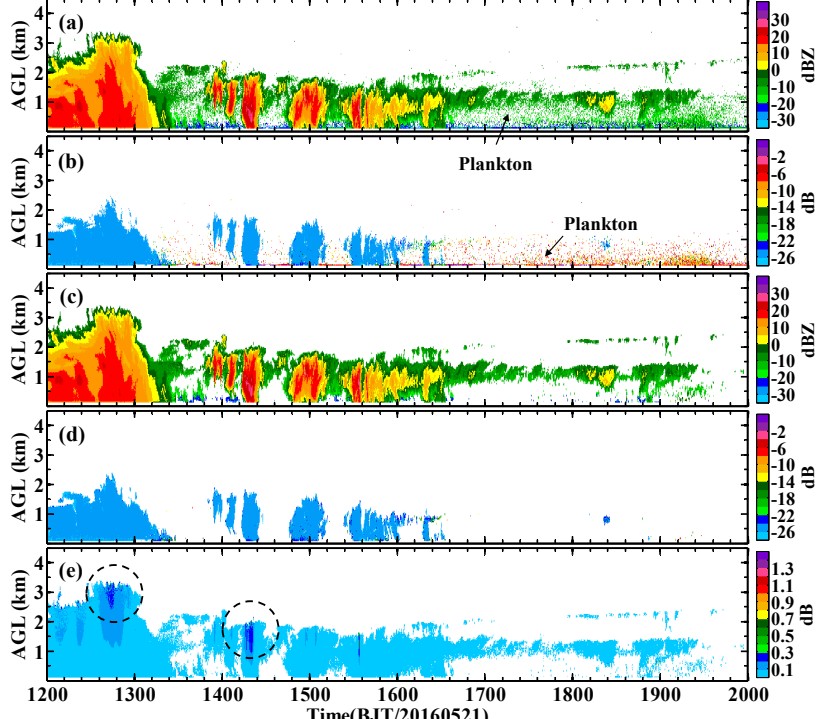

**Figure 5.** Time–height cross sections of the original reflectivity (**a**), original linear depolarization ratio (**b**), reflectivity after QC (**c**), linear depolarization ratio after QC (**d**), and the deviation (**e**) between (**a**) and (**c**), as observed by Ka-MMCR on 21 May 2016.

To illustrate the QC effect of PARSIVEL data, the measured diameter (D) and falling velocity ($V_f$) of raindrops in different classes were counted. Figure 6 shows the D-$V_f$ frequency distributions before and after QC. Upon comparing their differences, the raindrops greater than 1 mm, which possess a normal $D/V_f$ with an unrealistic $V_f/D$ and distribute far away from the theoretical curve, were eliminated after QC. Figure 7 presents the frequencies of the raindrops in different classes of D and $V_f$ before and after QC, and the comparisons show that some large raindrops with a D greater than 2.75 mm were unreliable measurements. Similarly, for $V_f$, the raindrops with a $V_f$ less than 1 m·s$^{-1}$ or greater than 13.6 m·s$^{-1}$ were also considered problematic data under abnormal circumstances, as described in Section 2.2.2.

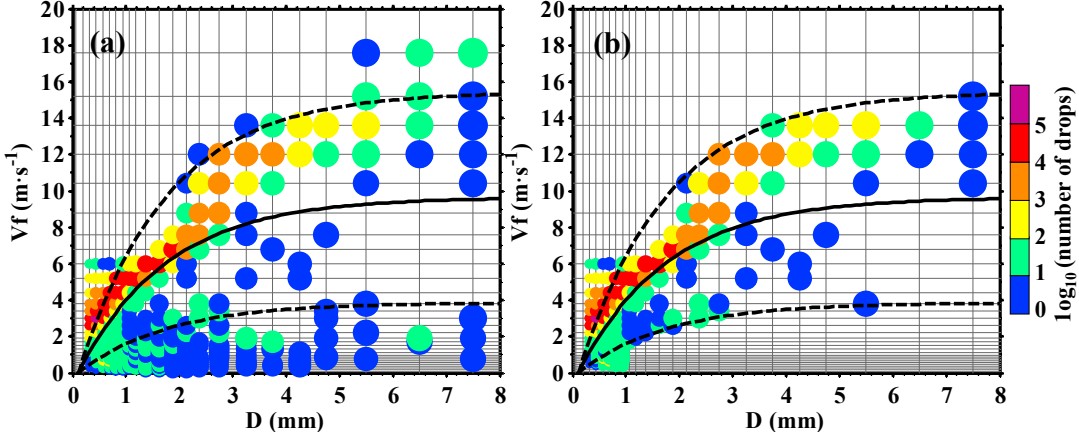

**Figure 6.** Raindrop diameter (D) -falling velocity ($V_f$) frequency distributions before (**a**) and after QC (**b**) for the entire PARSIVEL dataset. D represents the raindrop diameter, $V_f$ is the raindrop falling velocity, and the solid line represents the D-$V_f$ theoretical relationship under a still air condition.

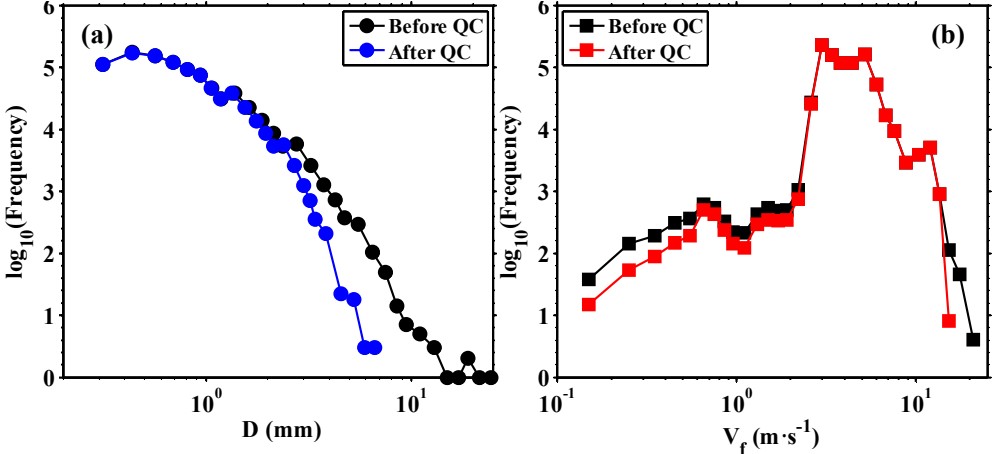

**Figure 7.** Frequencies of raindrops in different diameter (D) classes (**a**) and falling velocity ($V_f$) classes (**b**) before and after QC.

## 3.2. General Characteristics of the Hydrometeor Distribution

The datasets from Ka-MMCR and PARSIVEL were used to investigate the general characteristics of the hydrometeor distribution over and on the LM site during the observation period. Figure 8a shows the statistical result of the radar observation rates at different height levels (defined as the ratio of the sample number of warm clouds and precipitation to the total operational sample number). This shows that the majority of warm clouds and precipitation lie in the lower atmosphere under 4.2 km with radar observation rates gradually increasing as the altitude decreases, and 78.78% of the entire hydrometeors are concentrated below 2 km. Radar observation rates basically tend to be stable

within 0.84 and 0.39 km. Subsequently, they continue to increase to a maximum of 12.4% at 0.21 km. Figure 8b presents the PARSIVEL-measured accumulated rain durations and rain amounts under five $R_R$ regimes (0–0.1 mm·h$^{-1}$, 0.1–1 mm·h$^{-1}$, 1–5 mm·h$^{-1}$, 5–10 mm·h$^{-1}$, >10 mm·h$^{-1}$). The results show that the ground rainfall is dominated by light precipitation most of the time (confirmed that the radar echo was attached to the minimum detectable gate) with 85.26% of the entire duration having an $R_R$ smaller than 1 mm·h$^{-1}$. This part of the rainfall may be produced by weak cumulus and stratocumulus clouds. Only 14.74% of the entire duration had an $R_R$ greater than 1 mm·h$^{-1}$, which can be attributed to strong cumulus clouds and convection. Despite a large accumulated duration, light precipitation with a $R_R$ smaller than 1 mm·h$^{-1}$ only occupied 7.93% of the total rain amount. In contrast, the stronger precipitation with an $R_R$ greater than 1 mm·h$^{-1}$ produced 92.07% of the total rain amount.

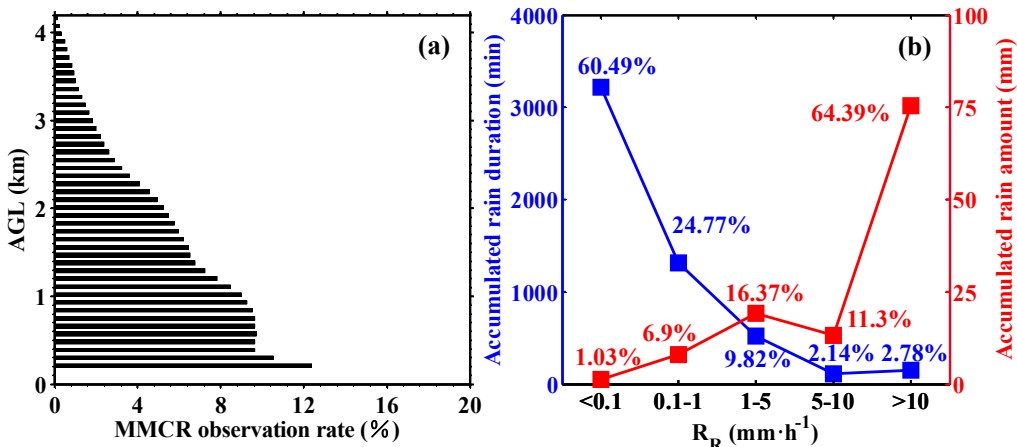

**Figure 8.** Ka-MMCR observation rates of warm clouds and precipitation at different height levels (**a**) and PARSIVEL-measured accumulated rain durations and rain amounts under five different rain rate ($R_R$) regimes (**b**).

### 3.3. Diurnal Variation of Warm Clouds and Precipitation

The matched datasets of 18 kinds of measurements and retrievals from the three instruments were further utilized to synthetically elucidate the diurnal variation of warm clouds and precipitation events. Note that only valid profiles containing cloud and precipitation targets of the Ka-MMCR and CL and rainy samples of the PARSIVEL were counted in the following figures and analysis.

The diurnal variations in CBH, CTH, and CTK for the entire events measured by Ka-MMCR and CL were shown in Figure 9. The statistics of CL-measured CBH (Figure 9a) show that 50% of the CBHs over 24 h were in a range of 0.21 to 2.13 km, and the remaining were scattered at different height levels due to the multilayered characteristic of warm clouds and precipitation in South China [46]. The CBHs from noon to the first half of the night were higher than that from the second half of the night to the forenoon, the mean CBHs during these two periods were in ranges of 1.049 to 1.171 km and 1.194 to 1.468 km, respectively. The latter period showed an average increase of 0.23 km of the CBHs. A gradually rising trend of the CBH during 1000–1900 BJT was found owing to the intensification of solar radiation. Comparing the CL-measured and Ka-MMCR-derived CBHs, the latter is perceived to be basically reliable. The radar-derived CBH also shows a rising trend from 1000 to 1700 BJT, and the bias of the averaged CBHs over 24 h for the two instruments was within −0.443 to 0.19 km (radar subtract CL). The boxplots also indicate that more than half of the CBHs were concentrated below the averaged CBH. The Ka-MMCR underestimates the CBH during 1800–2300 BJT and overestimates the CBH at 0800 BJT. Figure 9b,c presents the Ka-MMCR-derived CTH and CTK, respectively. They reveal that warm clouds and precipitation were shallow with 50% of the CTH and CTK over 24 h in ranges from 0.6 to 2.85 km and from 0.15 to 1.9 km, respectively. The CTHs and CTKs gradually increased from 1200 BJT to reach a maximum at 1600 BJT, and then gradually decreased until 2000 BJT.

The boxplots of CTH and CTK also suggested that more than 50% of the entire cloud cover was lower and thinner than the average values.

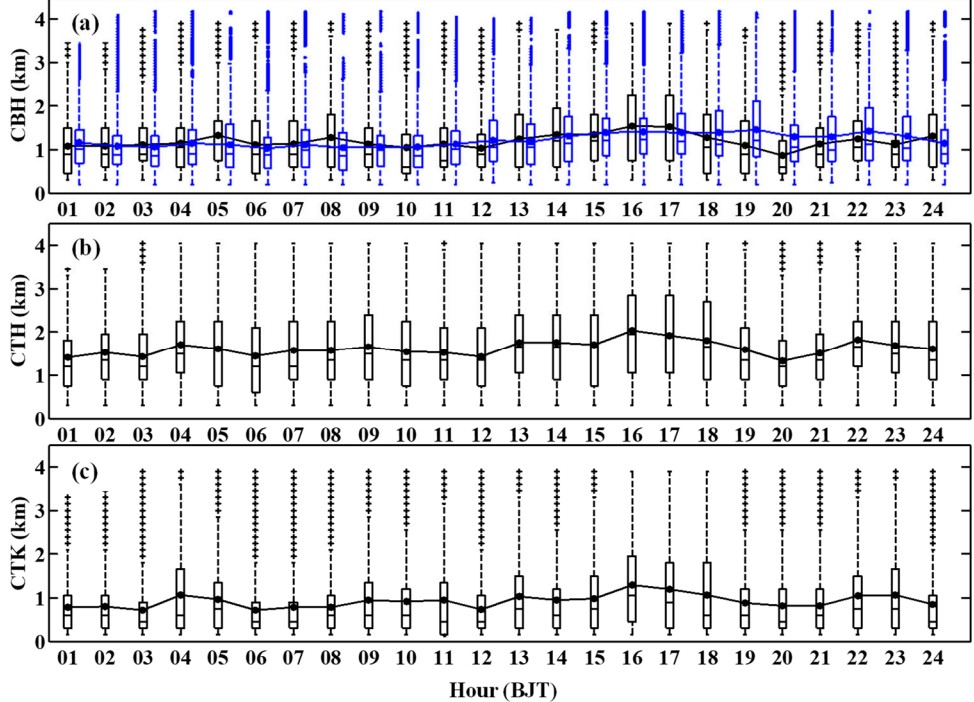

**Figure 9.** Diurnal variations in the cloud base height (CBH) (**a**), cloud top height (CTH) (**b**), and cloud thickness (CTK) (**c**) as measured by Ka-MMCR and CL during the observation period. The blue boxplot in (**a**) is the CL-measured CBH, and the others were measured by Ka-MMCR. On each box, the solid line with circles represents the average, the central cross-bar indicates the median, and the bottom and top edges of the box indicate the 25th and 75th percentiles, respectively. The dashed whiskers extend to the most extreme data points not considered outliers, and the outliers are plotted individually using the '+' symbol.

The diurnal time–height cross-sections of the average radar measurements and retrievals were analyzed to investigate the diurnal characteristics of the aloft warm clouds and precipitation events. The results of ($Z$, $V_M$, $S_W$, $S_K$, and $K_T$) and ($V_T$, $V_A$, $D_M$, $N_T$, LWC, and $R_M$) are shown in Figures 10 and 11, respectively. Note that only valid profiles with the existence of clouds and precipitation targets overhead were included in the calculation, and the time interval of the figures was set to ten min and the $Z$ was averaged in the logarithmic units. Figure 10a indicates that most of the warm clouds and precipitation was lower than 3 km, whereas a small part of the convections could develop up to 3.5 km. The radar reflectivities were uneven at different times; however, generally, the clouds and precipitation in the afternoon (1400–1800 BJT) and near midnight (2200–2300 BJT) were higher and stronger than at adjacent times, and this agrees with the variations in the CBH, CTH, and CTK. Some convections also appeared in the forenoon. The $Z$s of the entire events were within −20 and 17 dBZ. The $V_M$ (Figure 10b) was negative, implying the updraft was weak in the cloud body and the radar-returned Doppler information was dominated by the falling velocity of hydrometeors. The $S_W$ (Figure 10c) showed relatively large values exceeding 1 m·s$^{-1}$ near the cloud top, suggesting entrainment or particle growth processes were more frequent. The $S_K$ (Figure 10d) was small within −0.4 and 0.4 for the entire time–height image, indicating that physical processes in warm clouds are temperate, which can lead to formation of a symmetrical hydrometeor signal in the radar spectra [47]. However, the $K_T$ (Figure 10e) shows the difference between the cloud middle-upper part and the lower part. The cloud middle-upper part mostly exhibited a negative $K_T$, reflecting a mild cloud droplet growth process,

resulting in obtuse radar spectra. In contrast, the lower part of the cloud has a positive $K_T$ due to raindrop rapid collision and coalescence processes, resulting in peaky radar spectra [38,64].

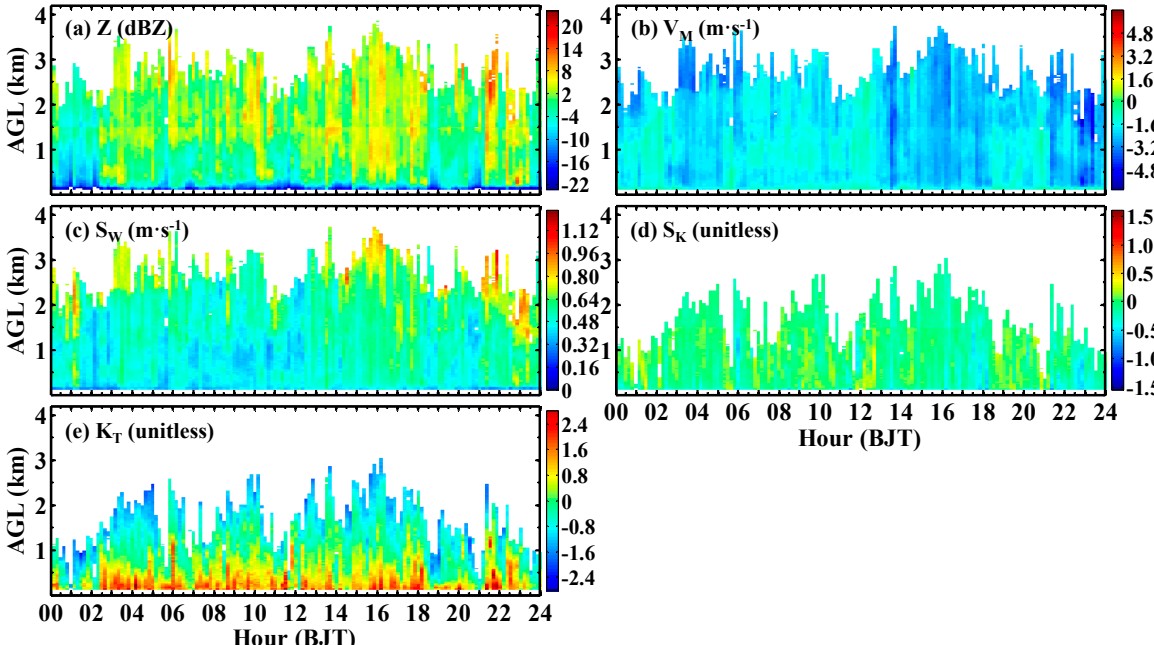

**Figure 10.** Diurnal time-height cross-sections of average radar-measured moments, reflectivity Z (dBZ) (**a**), mean Doppler velocity $V_M$ (M·s$^{-1}$) (**b**), spectrum width $S_W$ (m·s$^{-1}$) (**c**), spectral skewness $S_K$ (unitless) (**d**), and spectral kurtosis $K_T$ (unitless) (**e**).

The radar-derived $V_T$ (Figure 11a) concentrates in the range of −3 to 0 m·s$^{-1}$, which is much smaller than that in the deep convective and stratiform precipitations [47]. Some streaked structures with a relatively large negative $V_T$ appear in the image, and they concentrated more frequently during 1400–1800 BJT and 2200–2300 BJT than at other times, corresponding to frequent warm convections. Figure 11b–d shows that $V_A$ motions were very weak in these warm clouds and precipitation locations. The downdrafts and updrafts were within −3 and 0 m·s$^{-1}$ and 0 and 3 m·s$^{-1}$, respectively. The larger downdrafts appeared in a similar location with a larger Z because they were mainly induced by stronger convections. The updrafts mostly distributed in the low level, however, for some convections, they were larger and located in the middle and upper part of the cloud body. The $D_M$ and $N_T$ (Figure 11e,f) show that the mean diameter and number concentration of warm clouds and precipitation were in range of 0.1 to 0.5 mm and from $10^1$ to $10^5$ m$^{-3}$, respectively. Of note, the radar cannot detect the cloud droplets that are smaller than ~0.12 mm because of its limitation of sensitivity; therefore, the $D_M$ and $N_T$ are the result of large cloud droplets and raindrops. The $D_M$ and $N_T$ images indicate that the larger hydrometeors (0.25–0.5 mm) were mostly distributed in the low level below the CBH in low concentrations ($10^2$ to $3.16 \times 10^3$ m$^{-3}$). Conversely, small hydrometeors (0.1–0.25 mm) located above the CBH in high concentrations ($3.16 \times 10^3$ to $10^5$ m$^{-3}$). The LWC and $R_R$ images (Figure 11g,h) demonstrate that the warm clouds and precipitation on average possess liquid water and rain rates in the ranges of 0 to 0.5 g·m$^{-3}$ and 0 to 3 mm·h$^{-1}$, respectively. The streaked structures of LWC and $R_R$ present similar features as the $V_T$, and they both denote the existence of some strong warm convection events, which can produce larger raindrops that lead to a faster falling velocity, larger water content, and rain rate.

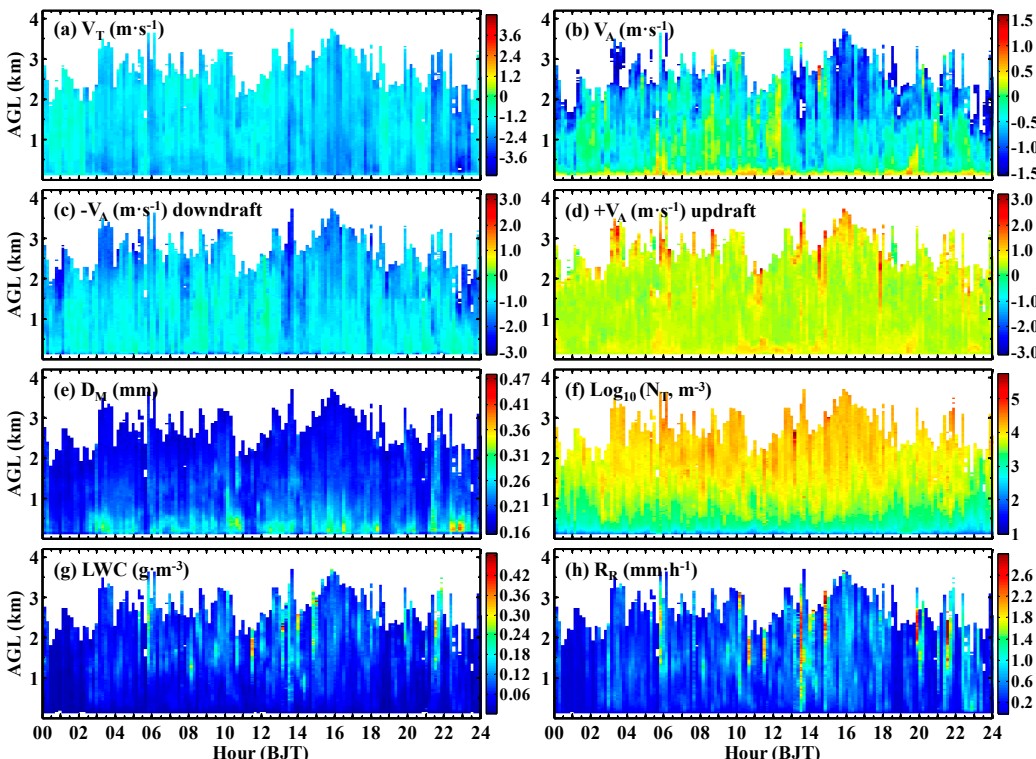

**Figure 11.** Diurnal time-height cross-sections of average radar-measured retrievals, particle mean terminal velocity $V_T$ (m·s$^{-1}$) (**a**), vertical air velocity $V_A$ (m·s$^{-1}$) (**b**), downdraft velocity (negative $V_A$) (**c**), updraft velocity (positive $V_A$) (**d**), particle mean diameter $D_M$ (mm) (**e**), particle total number concentration $N_T$ (m$^{-3}$) (**f**), liquid water content LWC (g·m$^{-3}$) (**g**), and rain rate $R_R$ (mm·h$^{-1}$) (**h**).

The ground-collected rainfall DSDs and other physical quantities were further determined to compare with the radar observations and complement the interpretation of the diurnal variation in warm precipitation. To obtain more representative results, the PARSIVEL measurements were calculated on an hourly interval to increase the statistical sample number. Figure 12 shows the diurnal variations in the hourly accumulated rain amount, accumulated rainy sample number, $R_R$, and average DSD, $D_M$, $N_T$, Z, and LWC. From the results of the eight parameters, we found that there were three periods of apparent rainfall during the entire day; they appeared at 0400–0700, 1400–1800, and 2300–2400. Specifically, the three periods produced higher rain amounts and rainy samples, which accounted for 83.23% and 56.01% of the entire accumulated rain amount and rainy sample number. However, the rain amounts and rainy samples at the other times were only 16.77% and 43.99% (Figure 12a,b). The raindrop spectra during these three periods were much wider with maximum diameters of raindrops approaching more than 3.5 mm. Meanwhile, they also had higher number concentrations (Figure 12d). The average $R_R$, $N_T$, and LWC (Figure 12c,f,h) also showed that the three periods had larger $R_R$s, $N_T$s, and LWCs than at the other times. For the three periods, the average values of $R_R$, $N_T$, and LWC were in the ranges of 1.1–2.13 mm·h$^{-1}$, $1.81 \times 10^2$–$2.54 \times 10^2$ m$^{-3}$, 0.057–0.126 g·m$^{-3}$; 0.5–1.57 mm·h$^{-1}$, $8.55 \times 10^1$–$1.51 \times 10^2$ m$^{-3}$, 0.027–0.09 g·m$^{-3}$, and 3.09–3.99 mm·h$^{-1}$, $1.66 \times 10^2$–$3.59 \times 10^2$ m$^{-3}$, 0.186–0.261 g·m$^{-3}$, respectively. For the other times, the average values of $R_R$, $N_T$, and LWC were in smaller ranges. The average values of $D_M$ and Z (Figure 12e,g) showed no evident diurnal variation, except during 2300–2400 BJT, and they were in ranges of 0.56–0.74 mm and 0.08–13.82 dBZ for the entire day. Upon comparing the averages and medians of $R_R$, $D_M$, $N_T$, Z, and LWC, we found that most of the medians were smaller than the averages, implying the majority of precipitation events had smaller values for these quantities than the averages. A small part of strong showers can contribute to larger proportions of rainfall quantities. The three apparent rainfall periods observed by PARSIVEL were basically in agreement with the radar result (Figure 10a), in that

the stronger rainfall on the ground was a response to the stronger echo overhead, corresponding to more frequent convective precipitation events. Interestingly, the raindrop concentration at 1300 BJT is much higher than that at the adjacent times, and possesses a smaller $R_R$ and LWC, indicating the rainfall during this period was mainly contributed by drizzle.

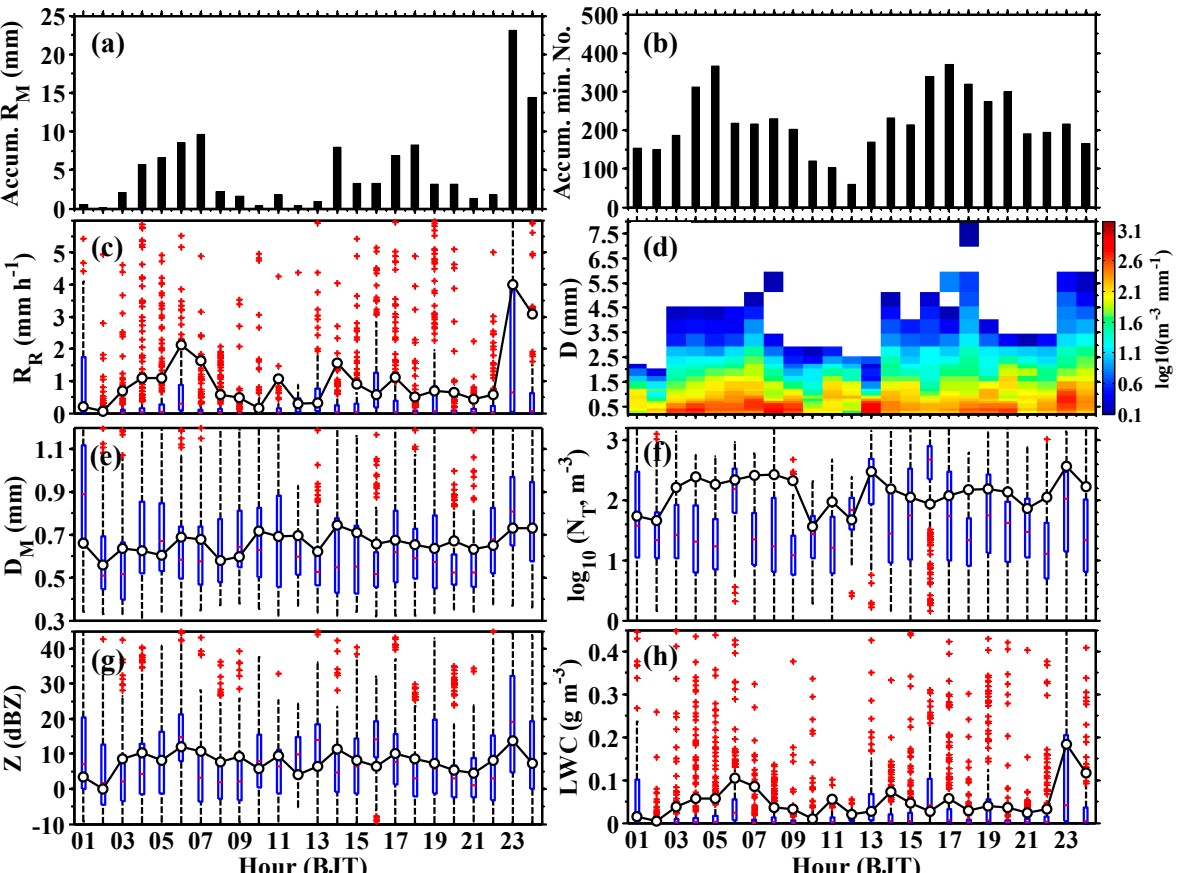

**Figure 12.** Diurnal variations in the hourly rainfall quantities measured by PARSIVEL, accumulated rain amount (**a**), accumulated minute number (**b**), rain rate $R_R$ (mm·s$^{-1}$) (**c**), average raindrop size distribution (**d**), particle mean diameter $D_M$ (mm) (**e**), total number concentration $N_T$ (m$^{-3}$) (**f**), reflectivity Z (dBZ) (**g**), and liquid water content LWC (g·m$^{-3}$) (**h**). The meanings of the symbols on the box are the same as described in the caption of Figure 9. Note that part of dash whiskers of the boxes which extend to the most extreme data points are not completely shown in order to present a better view of the features of the more representative dataset.

## 3.4. Vertical Structures of Warm Clouds and Precipitation

In this subsection, we discuss measurements from Ka-MMCR and CL that were calculated to further analyze the vertical structure of warm clouds and precipitation.

Figure 13 shows the occurrences of CBH, CTH, CTK, and CLN at different height levels as measured by the CL and Ka-MMCR. The CL results (Figure 13a) indicate that the CBH during occurrence of warm clouds and precipitation was in the range of 0.3 to 4 km, and the occurrence rapidly increased with height between 0.3 and 0.9 km to reach a maximum of 8.47%. However, afterwards, the occurrence gradually decreased as the height increased. Most of the CBH with an accumulated occurrence of 90.23% was below 2.2 km. The CTH presented in Figure 13b demonstrates that the MMCR-measured CTH was relatively divergent and distributed in the range of 0.6 to 4.2 km. The CTH occurrences at different heights were both nonnegligible with values from 1% to 5% because of the existence of various types of cloud and precipitation over the observation site. Considering the advantages and disadvantages of the MMCR and CL, the CTK is a synthetic result obtained by subtracting the

CL-measured CBH from the MMCR-measured CTH. The result (Figure 13c) shows a similar variation trend as the CBH in the range of 0.15 to 3.6 km. A majority of the CTK with 91.72% was thinner than 2.1 km, and a maximum of 13.8% was located at 0.6 km. For the cloud layer number, Figure 13d shows that 76.1% of CLN was single, and the occurrences of two-layer and three-layer CLN were 20.6% and 3.29%, respectively.

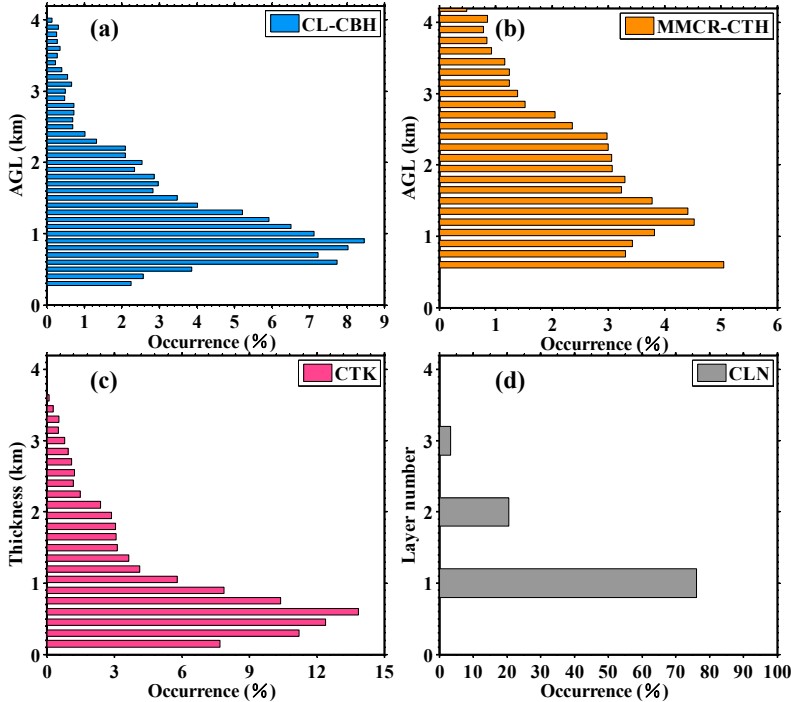

**Figure 13.** Occurrences of ceilometer (CL)-measured cloud base height CBH (**a**), Ka-MMCR-derived cloud top height CTH (**b**), synthetic cloud thickness CTK (**c**), and cloud layer number CLN (**d**), at different height levels.

Figure 14 presents the NCFADs (normalized contoured frequency by attitudes diagrams) of radar-measured Z, $M_V$, $S_W$, $S_K$, $K_T$, $V_T$, $V_A$, $D_M$, $N_T$, LWC, $R_R$, and CBH at different altitudes. The NCFAD is a kind of diagram that can conveniently express the frequency distribution of a quantity at different altitudes and value intervals [43]. Overall, Figure 14 shows two apparent features for the vertical distribution of warm clouds and precipitation. First, the value ranges of Z, $M_V$, $S_W$, $S_K$, $K_T$, $V_T$, $V_A$, $D_M$, and $N_T$ gradually became wider as the altitude decreased, indicating changes in the hydrometeor type and size in the vertical direction. Specifically, in the upper region, the hydrometeors mostly consisted of cloud droplets produced by non-precipitating clouds, whereas, as the altitude decreased, they gradually contained a greater proportion of raindrops yielded by precipitating clouds, exhibiting the occurrence of larger Z and $D_M$ values and smaller $V_M$, $V_T$, and $N_T$ values. Second, the majority of hydrometeors for the entire aloft warm clouds and precipitation events were located below 2.5 km, under which level they had apparent higher occurrences for all 12 quantities. The hydrometeors had a Z (Figure 14a) from −25 to 35 dBZ, most had values lower than 10 dBZ, corresponding to non- and light precipitating clouds; a small proportion can have stronger Z within 10 and 35 dBZ, which are produced by strong showers. The $V_M$ and $S_W$ (Figure 14b,c) mostly concentrate in ranges from −4 to 0 m·s$^{-1}$ and from 0.2 to 0.7 m·s$^{-1}$. A small part can reach up to −7.2 to −4 m·s$^{-1}$ and 0.7 to 1.25 m·s$^{-1}$. Figure 14d,e indicates that the contributions of small and large particles to radar signals are nearly symmetrical with small $S_K$s from −2.6 to 1.9, and as the altitude decreased, the radar spectra gradually became peaked with more raindrops. The $V_T$ (Figure 14f) gradually decreased with altitude, implying the formation of larger hydrometeors owing to the collision and coalescence processes of raindrops. However, as a feature of Z, the proportion of large raindrops with a larger particle falling velocity was

small. The vertical air motions, $V_A$ (Figure 14g), in the clouds and precipitation were in the range of $-3.4$ to 2.8 m·s$^{-1}$, and most were very weak within $-1$ and 1 m·s$^{-1}$. Relatively strong updrafts appeared in the low level under 0.66 km. The radar-derived $D_M$ and $N_T$ (Figure 14h,i) showed opposite features with height, namely, as altitude decreased, the hydrometeors gradually had larger mean diameters but lower number concentrations. However, this may not mean that the cloud layers were all precipitating, the CL-measured CBH distribution (Figure 14l) suggests that there were numerous cloud droplets at the low level; therefore, the vertical changes in $D_M$ and $N_T$ were caused by only a part of precipitation. Moreover, precipitation can exist within the cloud and can also cause the increase of $D_M$ and decrease of $N_T$ because of the raindrop growth processes, such as collision and coalescence. The LWC and $R_R$ (Figure 14j,k) for the entire warm clouds and precipitation were very small, with most values smaller than 0.15 g·m$^{-3}$ and 0.3 mm·h$^{-1}$, respectively, and a small part of strong showers had larger LWC and $R_R$ values that reached to maximums of 0.5 g·m$^{-3}$ and 1.8 mm·h$^{-1}$, respectively. Note that a horizontal line around 1.5 km in Figure 10, Figure 11, and Figure 14 was caused by the difference of the radar sensitivity around this altitude.

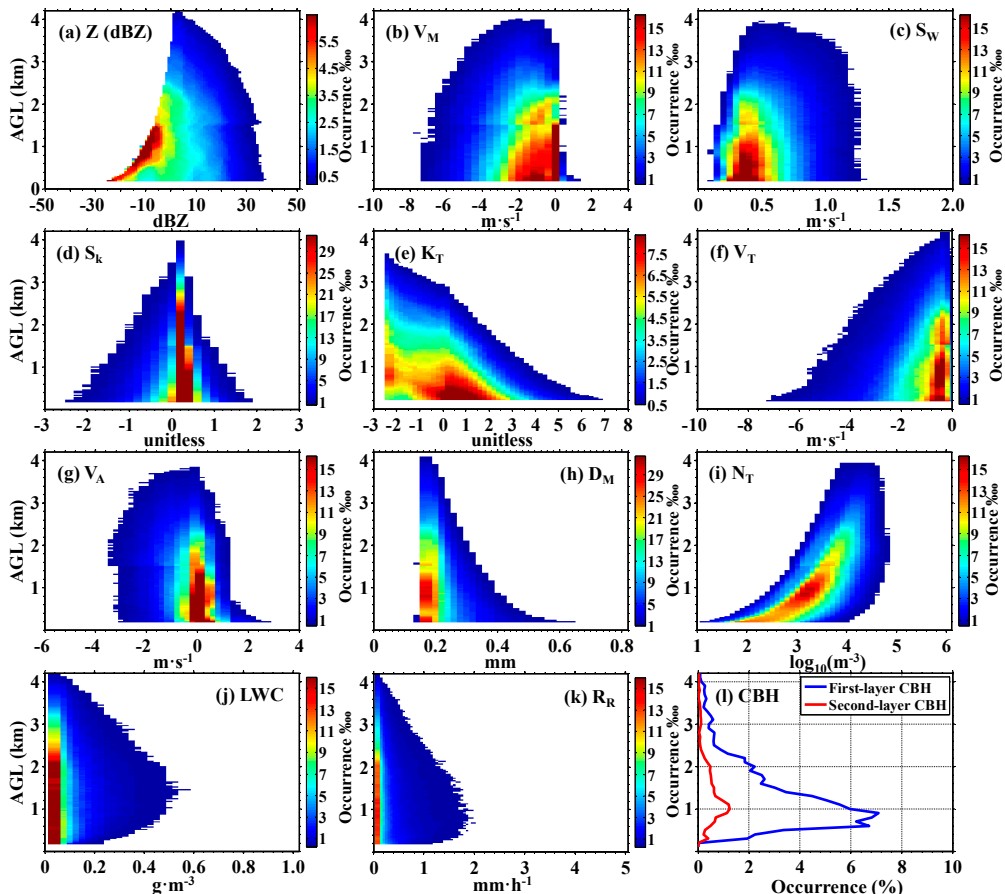

**Figure 14.** Normalized contoured frequency by altitude diagrams (NCFADs) of the Ka-MMCR-measured reflectivity Z (**a**), mean Doppler velocity $V_M$ (**b**), spectrum width $S_W$ (**c**), spectral skewness $S_K$ (**d**), spectral kurtosis $K_T$ (**e**), particle mean terminal velocity $V_T$ (**f**), vertical air velocity $V_A$ (**g**), particle mean diameter $D_M$ (**h**), particle total number concentration $N_T$ (**i**), liquid water content LWC (**j**), and rain rate $R_R$ (**k**) for the all warm clouds and precipitation events, and the probability distribution of the cloud base height (CBH) at different altitudes (**l**).

## 3.5. Raindrop Size Distributions of Warm Precipitation

Raindrop size distribution (RSD) is a vital piece of information for the parametrization of numerical models and the physical study of clouds and precipitation. The Ka-MMCR can simultaneously derive

a high-spatiotemporal resolution for the diameter and number concentration of hydrometeors from Doppler spectra. However, cloud droplets smaller than 0.12 mm, as shown in Figure 14h, were unavailable because of the limitation of radar sensitivity, implying the bulk of cloud droplets were not detected. Therefore, in this study, only the RSD of raindrops that typically have a diameter greater than 0.2 mm were calculated and analyzed. Figure 15 presents the average results of radar-retrieved and PARSIVEL-measured RSDs overhead and on the ground, respectively. The radar-retrieved RSD mainly covered the diameter range of 0.2 to 2.8 mm and in the concentration range of $2.265 \times 10^{-3}$ to $1.018 \times 10^4$ $m^{-3} \cdot mm^{-1}$, whereas the counterparts for PARSIVEL-measured RSD were from 0.312 to 5.5 mm and from $1.63 \times 10^1$ to $3.79 \times 10^3$ $m^{-3} \cdot mm^{-1}$. Thus, the RSDs for the warm precipitation are quite different in the air vs on the ground. To compare the observed RSDs, they were further fitted by utilizing a commonly used gamma distribution ($N(D) = N_0 D^\mu \exp(-\Lambda D)$) [65]. As shown in Figure 14 (the red lines), the fit results of the radar and PARSIVEL are very well with high linear correlation coefficients of 0.9998 and 0.9845, respectively. The three parameters of $N_0$, $\mu$, and $\Lambda$ were $1.49 \times 10^4$, $-0.9484$, and 6.79 for the radar (represent RSD overhead), and $1.875 \times 10^3$, 0.862, and 2.444 for the PARSIVEL (represent RSD on the ground).

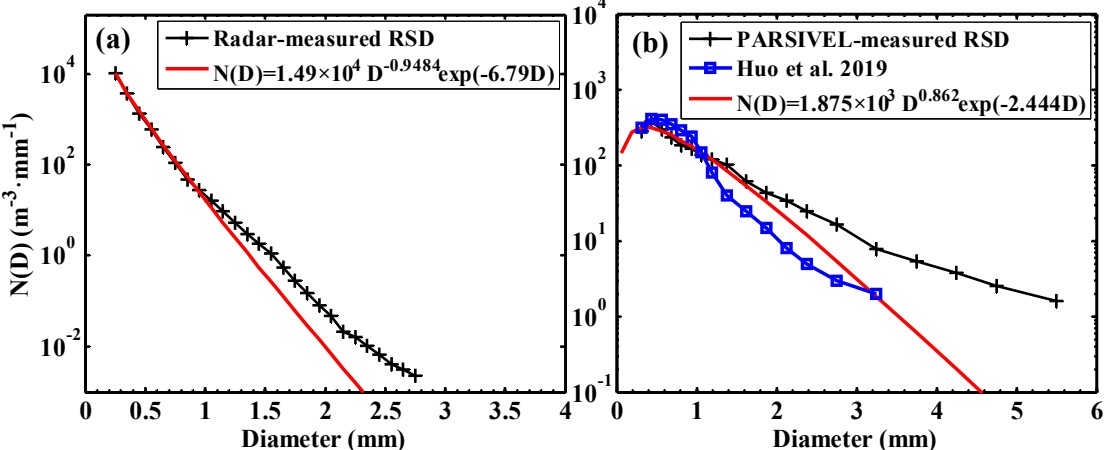

**Figure 15.** Average raindrop size distributions derived from the radar Doppler spectra (**a**) overhead and observed by PARSIVEL on the ground (**b**). The red lines are the fitted gamma distribution.

The different RSDs in the air and on the ground observed by the two instruments can be due to many reasons. One of the most important reasons is that the sampling targets of the two instruments are actually not synchronous in both time and space. The radar detected the raindrops in the air from 0.15 to 4.2 km, and within this altitude range, it can contain a proportion of raindrops that initially formed or have not suffered from the influences of physical processes, such as collision, coalescence, breaking, or evaporation. Therefore, the radar-derived RSD shows smaller diameters but higher concentrations than the counterparts observed on the ground. To further compare the results of two instruments under relatively fair circumstances, the retrievals from the radar were only counted under two limitations. First, they were only selected when PARSIVEL actually detected a valid $R_R$. Second, the result in the radar first range gate (0.15 km) was only used for comparison (to ensure a sampling height nearest to the ground). Based on these limitations, comparisons of the RSDs from the two instruments under different $R_R$s are shown in Figure 16. Thus, the biases of the two instruments are certainly different under the five regimes. Generally, the radar overestimates the concentrations of small raindrops, and it underestimates the concentrations of large raindrops. The overestimation can be attributed to the higher sensitivity of radar than that of PARSIVEL; in contrast, the underestimation can be caused by the attenuation and oversaturation of the radar signal. The boundaries for the overestimation and underestimation generally increase with $R_R$, and they were 0.5 mm, 0.6 mm, 0.75 mm, and 0.8 mm for the first four $R_R$ regimes (R1 < 0.1 $mm \cdot h^{-1}$, 0.1 ≤ R2 < 0.5 $mm \cdot h^{-1}$, 0.5 ≤ R3 < 1 $mm \cdot h^{-1}$, 1 ≤ R4 < 2 $mm \cdot h^{-1}$). When $R_R$ was greater than 2 $mm \cdot h^{-1}$, the samples were too rare to make a reasonable

comparison. Overall, the comparisons indicate that the biases between radar and PARSIVEL were relatively smaller for $R_R$ from 0.5 to 2 mm·h$^{-1}$ for raindrops with diameters of approximately 0.5 to 1 mm compared with the counterparts for other RR ranges for other sizes of raindrops.

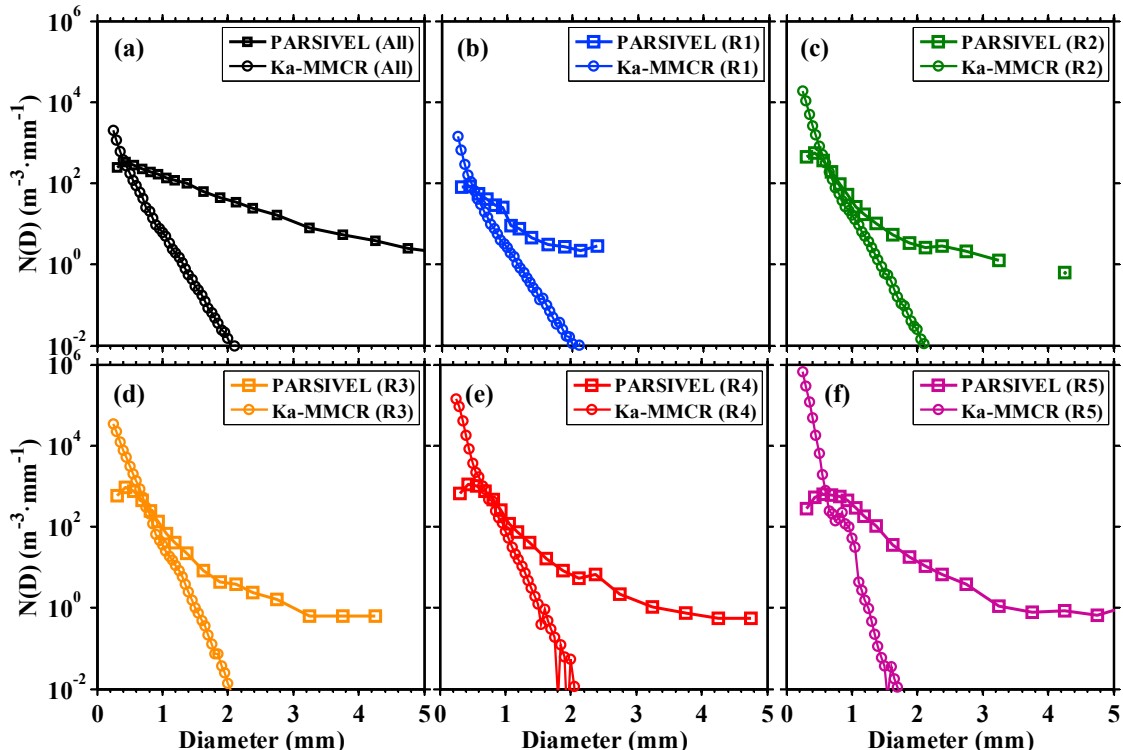

**Figure 16.** Comparisons of Ka-MMCR-retrieved (0.15 km aloft) and PARSIVEL-measured (on the ground) average raindrop size distributions of warm precipitation for five different $R_R$ regimes. (**a**) is the comparison for all measurements, (**b**)–(**f**) are comparisons for measurements under R1 to R5, which represent the five regimes of $R_R$s (R1 < 0.1 mm·h$^{-1}$, 0.1 ≤ R2 < 0.5 mm·h$^{-1}$, 0.5 ≤ R3 < 1 mm·h$^{-1}$, 1≤ R4 < 2 mm·h$^{-1}$, 2 ≤ R5 < 5 mm·h$^{-1}$).

## 4. Discussion

In this section, the findings in this study are further compared with previous studies and discussed according to the following aspects.

First, for the hydrometeors overhead and rainfall on the ground for the LM site during the pre-flood season, we found that the aloft hydrometeors observed by the Ka-MMCR and CL were mainly distributed below 3 km with a maximum occurrence of CBH near 0.6 km. This result is consistent with the study by Liu et al. in which the same instruments were used, but the datasets were collected during the summertime and the calculated hydrometeors overhead were limited below 3 km [2]. The ground rainfall observed by PARSIVEL showed that 85.26% of the entire warm precipitation events had a rain rate smaller than 1 mm·h$^{-1}$ to account for only 7.93% of the entire rain amount. In contrast, 14.74% of stronger precipitation with a rain rate greater than 1 mm·h$^{-1}$ can contribute to 92.07% of the entire rain amount. This feature is similar to the work by Huo et al. However, the latter focused on the precipitation during summertime, and the percentage of the duration for light shallow precipitation was 80% because small-scale precipitations with a duration less than 10 min were abandoned [3].

For the diurnal variation of warm clouds and precipitation, we found that the rainfall events mainly appeared during three periods of 0400–0700 BJT, 1400–1800 BJT, and 2300–2400 BJT, and the vertical distribution of hydrometeors were mainly located below 2.5 km with most of the radar reflectivities lower than 20 dBZ. This finding is quite different from the result over the tropical ocean area. For example, over the Southeast Pacific, warm precipitation events mainly occur before dawn

and usually disappear in the afternoon, and the hydrometeors were distributed below 4 km with most radar reflectivities greater than 20 dBZ [66].

Huo et al. also studied the RSD of shallow precipitation during the summertime for the LM site using a PARSIVEL distrometer, and the average RSDs are drawn in Figure 15 (the blue line). Upon comparing the result with ours (the black line), we observed that the two groups of RSDs are different, and the RSD of Huo has higher concentrations of small raindrops with diameters from 0.312 to 1.062 mm. In contrast, it possesses lower concentrations of large raindrops with diameters from 1.187 to 3.25 mm. This difference can be attributed to two main reasons. The first reason is the differences in the PARSIVEL data processing. All samples at a $R_R$ smaller than 0.1 mm·h$^{-1}$ were removed, and a size correction procedure was implemented in Huo's study. The second reason is the difference in observation periods. The data used in that study were collected during the summertime, which may have led to different statistical characteristics of warm precipitation compared with our results during the pre-flood season.

## 5. Conclusions

In this study, we developed integrated technology for using a Ka-band MMCR, CL, and disdrometer for investigation of the vertical structure, diurnal variation, and physical properties of warm clouds and precipitation. This technology was based on the advantages of each instrument and appropriate data processing and QC methods. The technology was implemented to study the warm clouds and precipitation in South China during the pre-flood season in 2016. The main conclusions can be addressed as follows.

The warm clouds and precipitation over this region were mainly distributed in low altitudes with 90% below 2.5 km. Most of the warm rainfall events were very light, and 85.26% had a $R_R$ under 1 mm·h$^{-1}$ to account for only 7.93% of the entire rain amount. In contrast, 14.74% of the stronger precipitation contributed to 92.07% of the entire rain amount. More than half of the CBH, CTH, and cloud thickness values were in the ranges from 0.21 to 2.13 km, 0.6 to 2.85 km, and 0.15 to 1.9 km, respectively. There was a rising trend of CBH from 1000 to 1900 BJT, and the CTH and cloud thickness gradually increased from 1200 BJT to reach a maximum at 1600 BJT and then decreased until 2000 BJT. Additionally, 76.1% of the cloud layers were singular, and double and triple layers accounted for 20.6% and 3.29%, respectively. There were three periods of apparent rainfall on the ground during the day, namely, 0400–0700 BJT, 1400–1800 BJT, and 2300–2400 BJT. During the three periods, the accumulated rain amount accounted for 83.23% of the entire rain amount, and the raindrops had wider size spectra, a higher number concentration, larger rain rates, and a higher water content than the counterparts at the other times. Relevant apparent precipitation in the air observed by the radar occurred during two periods of 1400–1800 BJT and 2200–2300 BJT, and the first period was the same as the result on the ground, whereas, the second period was one hour earlier. It seems that the radar misses more rainy measurements during 0400–0700 BJT than during those two periods. During these two periods, the radar-observed reflectivity was stronger, particle falling velocity was faster, and a large $R_R$ was more concentrated than at the other times. In the vertical orientation, the hydrometeor type, size, and concentration were gradually changed as the altitude decreased. In the upper region, the hydrometeors mostly consisted of cloud droplets, whereas, as the altitude decreased, they gradually contained a higher proportion of raindrops resulting from precipitating clouds, exhibiting the occurrence of a part of larger Z and $D_M$ values and smaller $V_M$, $V_T$, and $N_T$ values. Comparisons of the radar-derived RSD at 0.15 km above the site and the disdrometer-measured RSD on the ground showed that the RSDs were quite different, and their biases were relatively smaller in the $R_R$ range of 0.1 to 2 mm·h$^{-1}$ for raindrops with diameters approximately from 0.5 to 1 mm compared with the counterparts in the other RR ranges for other sizes of raindrops. Inevitably, the asynchronous sampling in both time and space of the instruments and their individual limitations can also induce uncertainties in the comparison. Gamma distributions for the raindrop size distributions in the air and on the ground were fitted to be

N(D) = 1.49 × 10$^4$D$^{0.9484}$exp(−6.79D) and N(D) = 1.875 × 10$^3$D$^{0.862}$exp(−2.444D), where D and N(D) are the diameter and number concentration of the raindrops.

**Author Contributions:** Conceptualization, J.Z. and L.L.; methodology, J.Z.; software, J.Z., H.C.; validation, H.C. and Y.G.; formal analysis, Y.C.; investigation, J.Z.; resources, L.L.; data curation, J.Z. and L.L.; writing—original draft preparation, J.Z.; writing—review and editing, Y.C., H.C., and Y.G.; visualization, Q.L.; supervision, H.X.; project administration, J.Z.; funding acquisition, J.Z.

**Funding:** This research was funded by the Major Research Plan of the National Natural Science Foundation of China (Grant No. 91537214), the National Natural Science Foundation of China (Grant Nos. 41705008, 41905084), and the Scientific Research Foundation of Chengdu University of Information Technology (Grant No. KYTZ201728).

**Acknowledgments:** The authors would like to thank the Chinese Academy of Meteorological Sciences for providing the radar data. Thanks also go to the reviewers for thorough comments that really helped to improve the manuscript.

**Conflicts of Interest:** The authors declare no conflict of interest.

**Appendix A**

**Table A1.** Abbreviations used in this paper.

| No. | Abb. | Meaning | No. | Abb. | Meaning |
|-----|------|---------|-----|------|---------|
| 1 | MMCR | millimeter-wave cloud radar | 13 | $K_T$ | spectral kurtosis |
| 2 | SP | Doppler spectrum | 14 | CTK | cloud thickness |
| 3 | Z | reflectivity | 15 | CLN | cloud layer number |
| 4 | $V_M$ | mean Doppler velocity | 16 | $V_A$ | vertical air velocity |
| 5 | $S_w$ | spectrum width | 17 | $V_T$ | particle mean terminal velocity |
| 6 | LDR | linear depolarization ratio | 18 | $D_M$ | particle mean diameter |
| 7 | CBH | cloud base height | 19 | $N_T$ | particle total number concentration |
| 8 | CTH | cloud top height | 20 | LWC | liquid water content |
| 9 | DSD | drop size distribution | 21 | LM, | Longmen Weather Observatory |
| 10 | $R_R$ | rain rate | 22 | $V_f$ | falling velocity |
| 11 | $R_M$ | rain amount | 23 | QC | quality control |
| 12 | $S_K$ | spectral skewness | 24 | BJT | Beijing Standard Time (UTC+8) |

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
