# Peer review of "Characteristics of Warm Clouds and Precipitation in South China during the Pre-Flood Season Using Datasets from a Cloud Radar, a Ceilometer, and a Disdrometer"

_remotesensing, doi:10.3390/rs11243045_

Round 1

Reviewer 1 Report

Please see attached review.

Author Response

Dear reviewer,

Thank you very much for your careful reading and constructive comments on our manuscript, which are all valuable and very helpful for revising and improving our manuscript. We have made every effort to revise the manuscript according to your comments and suggestions.

We appreciate your kind work.

Best regards,

Jiafeng Zheng and the co-authors

Reviewer 2 Report

Review – A study of characteristics of Warm Cloud-precipitation in South China […] Using Datasets Observed from a Cloud Radar, Ceilometer, and Disdrometer

The scientific basis of this work is deep and sound, and I think that just the methodology of this work would be enough to really deserve publication. However, I recommend a number of changes before this, since I feel that the paper needs some fundamental restructuring and rewriting.

First of all, there is not a true direction to the paper. Starting with the abstract, we get a number of descriptions of a warm cloud precipitation which is really failing to illustrate the scope of the work, and seems to point to nothing more than a case study. Both in the abstract and in the paper, the aim of the study is not stated clearly; at the end of the introduction, the authors say that the main goal of the paper is to “investigate the diurnal variation, vertical structure [...] of warm cloud-precipitation in South China” which would make it nothing more than a case study of applying instruments and measure;  so there is no clear indication to why such a work should interest the reader and deserve publication.

What distinguishes this paper from being a mere case study of precipitation over China with various instruments, something that wouldn’t make it worthy of publication, are the data processing and quality control techniques and the combined use of several methods of investigation. The methodology part of the paper is especially interesting and should be much emphasized in the Abstract, while it is not mentioned at all. The reason for combining the methods, should flow naturally enough from the limitations of individual methods pointed out in the Introduction, but they are left implicit in the text.

For instance, formulas of particle size distributions for warm cloud and precipitation in different reflectivity regimes are proposed, and this is a result of the methods employed and possibly a major finding of the study. It is not mentioned in the Abstract nor emphasized anywhere in the text as a possible goal before it pops up in the Results.

Moreover, the Discussion part of the paper is not really a Discussion. The Discussion should put the results in context with other studies investigating similar and/or related phenomena and instrumentation used, while currently the Discussion part seems mostly to be an extension of the Results.

The quality of the English used in the paper is good enough to have a proper understanding of the subject matter, of the methods used and of the results, but it can and should be improved. Many sentences are too long and cumbersome to read. A revision from a native speaker is suggested, but I have tried to point out sentences and bits that do not feel right to my ears. I would also like to point out that I am not a native speaker so I really encourage the authors to seek some assistance for language review.

Detailed comments

Abstract.

Lines 19-22 “Warm cloud-precipitation...” to “numerical models” feels a bit redundant. In my opinion it might feel better as a part of an introduction than an abstract.

I would re-write along these lines:

“In this study, we developed a method to integrate different measurement techniques into the study of characteristics of warm cloud-precipitation, in order to advance our understanding of its internal physical processes. Three instruments, a Ka-band MMCR, a ceilometers, and a disdrometer, were simultaneously deployed at Longmen weather observatory of Guangdong Province, in order to provide continuous, long-term and high-resolution measurements of cloud-precipitation over the site. A mix of novel and standard data processing, quality control and retrieval techniques have been used  to improve data quality and obtain additional properties of cloud-precipitation. [etc etc, e.g., alleviate signal attenuation of MMCR, correct overestimation/remove problematic data of disdrometer]”

Line 35-36: “that they owned different vertical structures and physical properties” - - > “that they possessed”

I would add in the abstract  something about particle size distributions of warm cloud and precipitation (Section 3.5) - - > “formulas of particle size distributions for warm cloud and precipitation in different reflectivity regimes were proposed” or something along these lines.

Introduction

I am not an expert on instrumental techniques, so it is difficult for me to fully determine how sufficient and extended the references presented here are. More specific comments follow.

Line 47: “low level of atmosphere” - - > “the atmosphere”

Line 50-54: overlong sentence, should be broken down (moreover, not sure what “exceptional detection” is). It can be changed into something along these lines - - > Detection and study of warm cloud-precipitation by remote sensing can be valuable to improve our understanding of its physical and dynamic issues. They can provide necessary information for numerical weather and climate models, and help in conducting realistic missions of weather modification”.

Lines 55-57: “with certain purposes under specific scientific backgrounds” - - > vague, can be left out, since it is specified later on in the paragraph what the various methods are used for.

Line 74-75: “They can provide data without any topographic limitations, especially over oceans and plateaus, where are difficult for other technologies to achieve” - - > replace second part, it’s not correct in English.

Line 79 - - > leave out “et al”

Line 81 “the request of the” - - > I wonder whether “requirements of the” is better?

Line 86 “when they equip” - - > maybe “when they are equipped”?

Line 93 - - > cut out “and et al.”

Line 93-97 - - > re-write – sounds wrong in bits like “applied in many atmospheric science fields”, plus “evaluating the cloud radiative effect/investigating features of high-impact severe weather systems, etc.” are not really a science “field”, maybe “topic” or “studies” or some other word is better suited

Line 98, “there are some deficiencies of those radars for” - - > “they possess some limitations in”

Line 99, “with limited times per day” - - > “a limited number of times per day”

Line 101, “in low altitude” - - > “at low altitudes”

Line 111 “because of the scattered ability of a hydrometer is inversely proportional..:” - - > something wrong in this sentence

Lines 146-149 - - > reformulate the main purpose of the paper. Investigating properties of warm cloud-precipitation is a case study, not the focus of a research article. There is more than enough interesting material in this paper to make it worthwhile for a research paper publication. For instance, it could be replaced by something like: “The main purpose of this paper is to develop an integrated method for using three different measurement techniques (Ka-band MMCR, laser ceilometers, and disdrometer) into the investigation of diurnal variation, vertical structure, and macro- and microphysical properties of warm cloud-precipitation. This method has been applied to precipitation in South China during the pre-flood season in 2016...”

Various minor English problems pop-up here and there throughout the paper.

Materials and Methods

I am not an expert on technical matters, but the explanations seem quite clear to me and well-structured. A few English problems here and there popup, just like I said above.

Results

As we have discussed about a re-designing of the paper goals, I would maybe introduce the results (or end the results) with a presentation/summary/discussion of how using different techniques improved the general characterization of precipitation. There are a lot of figures and results so a way to navigate among them might also be useful, something like "The results section is structured in this way: etc etc"

Figures and Tables

Figure 1. Yellow text not clear.

Table 1 and Table 2. No clear separation between table sections – where does radar system vs radar precipitation mode stop and begin? The same about ceilometers and disdrometer, not all readers are familiar with the instruments and the read is difficult without a space or separation line

Figure 3 seems to have some problem, blurred numbers (compare it with figure 4 just below)

Figures 10-11. name of variables on the legend on the right is very very small and difficult to read when paper printed. Sometimes we forget that our colleagues not always have availability of laptop/pc where we can zoom in figures at will!

Figure 14. Too many figures crammed in too little space. Legend on the right almost undreadable when paper printed.

-------------------------------------------------------------------------------------

Last but not least, I would like to express my gratitude to the Editor and Authors for the interesting read and for the opportunity to discuss and improve on this work.

I wish the Authors best luck in their quest for publication and I am looking forward to reading the improved paper.

Kind regards

Author Response

(The authors gave the same response as above.)

Round 2

Reviewer 1 Report

Dear authors,

Thank you very much for your comprehensive and detailed responses to my review comments. I really appreciate the effort that has been put in to respond carefully to each point. The paper is much improved and will be ready for publication after a few simple minor notes are considered.

These minor comments are:

Line 30: "within 1" should be "below 1" or "less than 1".

Line 41: "raindrop" -> "raindrops"

Line 341: PASERVEL -> PARSIVEL

I appreciate that the authors have removed the distinction between cloud and precipitation in their results. However some of the discussion still draws a distinction that I think is misleading. For example on line 131, the "raindrop size distribution of warm clouds and precipitation" is mentioned, but by definition the raindrop size distribution is for precipitation and not clouds. So the paper should be carefully checked to ensure that the raindrop size distribution refers to precipitation only and that the definitions of warm cloud and precipitation properties are carefully made.

Line 339: By "precipitating profile" I guess the authors mean profiles that contained precipitation all the way to the ground, not only aloft? This is unclear here.

Line 305: "has a normal" -> "that has a normal"

I thank the authors for updating the sampling area calculation. As I mentioned in my review, the choice of formula depends on whether edge fallers are removed by the instrument. If edge fallers are automatically removed, as in newer Parsivels, the formula should use D instead of 0.5D. Whether or not edge fallers are automatically removed is a function of the age of the instrument.

Line 318: In the equation for the sampling area the drop diameter is in mm but the width/height of the beam are in m; please correct and ensure the calculation is with the correct units.

Line 365: "remains well" -- meaning "remains"?

Line 405: not all drops greater than 2.75 mm were removed. I suggest writing "some large raindrops with a D greater than 2.75".

Line 402: The choice of 1 mm as a threshold here should be justified in the manuscript (it is already justified in the responses to reviews).

Line 448: "increase 0.23 km" -> "increase of 0.23 km". The authors should state whether this difference a difference in means, and whether it is statistically significant, given the large scatter in the CBH values.

For boxplot captions, the different symbols should be explained (which is median, which is mean etc).

Line 476: "no apparently altitude distribution" - does this mean no apparent changes with altitude?

Lines 478-486: Please include some references for the microphysical interpretations presented here.

I thank the authors for the explanation of the horizontal line in the figures 10, 11, and 14. I think this explanation should be included also in the manuscript, as a brief note.

Line 488: "small" -> "smaller"

Line 502: Fig 11. g is labelled RWC but here is referenced as LWC.

Figure 12: Do the vertical dashed lines indicate that there are points off the plots that exist but are not shown here? If so an explanation should be included. Again the symbols need an explanation in the caption.

Figure 12: It appears that the drop concentration was higher around 13h than in the "peak" rainfall periods. Is this due to drizzle, and can the authors add a comment?

Line 542: I would suggest writing "more frequent", since the results do not show particularly strong convection for these times, but rather a higher incidence of convective features.

Lines 552-553: Precipitation can not have a CBH because it is liquid water and not cloud; I suggest rephrasing to write "CBH during occurrence of warm clouds and precipitation".

Line 572: Please include a reference for NCFAD.

Line 579: The values responded to the changes and not the other way around; please rephrase.

Figure 14: The caption has M_V instead of V_M for panel b.

Line 597: Figure 14k -> 14i for CBH distribution.

Line 596: Precipitation can exist within the cloud, so the fact that the cloud base was often low does not remove the possibility of changes in Dm and Nt being due to precipitation processes. Increasing Dm and decreasing Nt would normally indicate raindrop growth (collision and coalescence), including within the cloud. This should be mentioned as a possibility here.

Line 609: "Particle" -> "Raindrop" (to match "RSD")?

Line 617: "changed in the" -> "covered the"

Line 620: "determine" -> "compare"?

Line 623: The fit quality is not shown particularly by these log diagrams. I would simply state the linear correlation coefficients for each fit.

Figure 16: It would be better to show this figure with large x and y axes so the points are not truncated and the full DSDs are shown.

Line 643: "PARSERVAL" -> "PARSIVEL"

Line 648: "too rare that can effect the comparison" -> "too rare to make a reasonable comparison"

Line 649: This conclusion is confusing. I see the best match for drops between about 0.5 to 1 mm when the rain rates are between 0.5 and 2 mm/h. This should also be updated on Line 718.

For ranges that overlap (like 0.1 to 0.5, 0.5 to 1), the authors should indicate which end of the interval is open and which is closed.

Line 660: "mainly" -> "were mainly"

Line 663: What is meant by "statistical hydrometeors"?

Line 667: "later" -> "latter"

Line 675: "distribute" -> "are distributed"

Line 697-698: These percentage values do not all match with those on lines 665-666, and these also do not all match with earlier values quoted on lines 432-433.

Line 703: % missing after 3.29.

Line 709: Did the radar then miss the more rainy period from 0400-0700?

Line 713: Given the minimum detectable drop size was 0.12 mm, perhaps it would be more correct to remove "produced by non-precipitating clouds"?

Line 714: "results" -> "result"

Line 715: Again the physical processes produce changes in the variables, not the other way around.

Line 716: "aloft" -> "above"

Author Response

Dear reviewer,

Thank you very much for your careful reading and detailed comments, your kind work is invaluable for improving our manuscript. We have revised the manuscript according to your comments and suggestions again.

Thank you for your kind work again.

Best regards,

Jiafeng Zheng and the co-authors
